# Spatial and temporal variations in glacier aerodynamic surface roughness during the melting season, as estimated at August-one ice cap, Qilian mountains, China

5    Junfeng Liu[1]*, Rensheng Chen[1], Chuntan Han[1]

[1] Qilian Alpine Ecology and Hydrology Research Station, Key Laboratory of Ecohydrology of Inland River Basin, Northwest Institute of Eco-Environment and Resources, Chinese Academy of Sciences, Lanzhou, China.
*Correspondence to: Rensheng Chen (crs2008@lzb.ac.cn), Junfeng Liu (jfliu121@163.com)*

**Abstract:** The aerodynamic roughness of glacier surfaces is an important factor governing turbulent heat transfer. Previous studies rarely estimated spatial and temporal variation in aerodynamic surface roughness ($z_0$) over a whole glacier and whole melting season. Such observations can do much to help us understand variation in $z_0$ and thus variations in turbulent heat transfer. This study, at the August-one ice cap in the Qilian mountains, collected three-dimensional ice surface data at plot-scale, using both automatic and manual close-range digital photogrammetry. Data was collected from sampling sites spanning the whole ice cap for the whole of the melting season. The automatic site collected daily photogrammetric measurements from July to September of 2018 for a plot near the center of the ice cap. During this time, snow cover gave place to ice and then returned to snow. $z_0$ was estimated based on microtopographic methods from automatic and manual photogrammetric data. Manual measurements were taken at sites dotted from terminals to top; they showed that $z_0$ was larger at the snow and ice transition zone than in areas fully snow or ice covered. This zone moved up the ice cap during the melting season. It is clear that persistent snowfall and rainfall both reduce $z_0$. Using data from a meteorological station near the automatic photogrammetry site, we were able to calculate surface energy balances over the course of the melting season. We found that high or rising turbulent heat as a component of surface energy balance tended to produce a smooth ice surface and a smaller $z_0$; low or decreasing turbulent heat tended to produce a rougher surface and larger $z_0$.

**Keywords**: aerodynamic surface roughness, digital photogrammetry, melting season, transition zone, surface energy balance, August-one ice cap

## 1. Introduction

The roughness of ice surfaces is an important control on air-ice heat transfer, on the ice surface albedo, and thus on the surface energy balance (Greuell and Smeets, 2001; Hock and Holmgren, 2005; Irvine-Fynn et al., 2014; Steiner et al., 2018). The snow and ice surface roughness at centimeter and millimeter scales is also an important parameter in studies of wind transport, snowdrifts, snowfall, snow grain size, and ice surface melt (Bruce and Smeets, 2000; Brock et al., 2006; McClung and Schaerer,

2006; Fassnacht et al., 2009a; Fassnacht et al., 2009b). Radar sensor signals, such as SAR (Oveisgharan and Zebker, 2007), altimeters and scatter meters, are also affected by ice and snow surface roughness (Lacroix et al., 2007; Lacroix et al., 2008). One of the most important of these influences is the aerodynamic roughness of $z_0$, which is related to ice surface topographic roughness in a complex way (Andreas, 2002; Lehning et al., 2002; Smith et al., 2014; Smith et al., 2016). Determination of $z_0$ based on topographic roughness is therefore of great interest for energy-balance studies (Greuell and Smeets, 2001).

Glacier surface $z_0$ has been widely studied through such methods as eddy covariance (Munro, 1989; Smeets et al., 2000; Smeets and Van den Broeke, 2008; Fitzpatrick et al., 2019), or wind profile (Wendler and Streten, 1969; Greuell and Smeets, 2001; Denby and Snellen, 2002; Miles et al., 2017; Quincey et al., 2017). However, micro-topographic estimated $z_0$ shows some advantages, such as    lower scatter, rather than profile measurements over slush and ice (Brock et al., 2006), and ease of application at different locations (Smith et al., 2016). Current research has increasingly used micro-topographic method to

estimate $z_0$. It has also become clear that it is important to estimate $z_0$ over the entire course of the melting season and at many points on the glacier surface, as $z_0$ is prone to large spatial and temporal variation (Brock et al., 2006;Smeets and Van den Broeke, 2008). This variation is due to variations in weather and snowfall (Albert and Hawley, 2002). The micro-topographic estimated $z_0$ allows repeated measurement at many points on the glacier surface, which is not possible with wind profile or eddy covariance methods.

Photogrammetry has been increasingly popular as a method to measure the aerodynamic surface roughness of snow and ice(Irvine-Fynn et al., 2014; Smith et al., 2016; Miles et al., 2017; Quincey et al., 2017; Fitzpatrick et al., 2019).. Initially, the Micro-topographic method was developed as snow digital photos were taken against a dark background plate. The contrast between the surface photo and the plate could then be quantified as a measure of glacier roughness (Rees, 1998). This methods still widely applied to quantify glacier surface roughness (Rees and Arnold, 2006; Fassnacht et al., 2009a;Fassnacht et al.,

2009b;Manninen et al., 2012). A more recent method, as described by Irvine-Fynn et al. (2014), uses modern consumer-grade digital cameras to do close-range photogrammetry at plot scale (small plots of only a few square meters). Appropriate image settings and acquisition geometry allow the collection of high-resolution data (Irvine-Fynn et al., 2014; Rounce et al., 2015; Smith et al., 2016; Miles et al., 2017; Quincey et al., 2017). Such data facilitates the distributed parameterization of aerodynamic surface roughness over glacier surfaces (Smith et al., 2016; Miles et al., 2017; Fitzpatrick et al., 2019). Precision

of microtopographic estimated $z_0$ also became a major concern, and many comparative studies with the aerodynamic method (eddy covariance or wind towers measurements) were carried out over debris-covered or non-debris covered glaciers. The difference was within an order of magnitude for some studies (Fitzpatrick et al., 2019) or strongly correlated (Miles et al., 2017).

Previous researchers have performed some long-term, systematic studies of glacier surfaces (Smeets et al., 1999; Brock et al.,

2006; Smeets and Van den Broeke, 2008; Smith et al., 2016). The current study applied such methods to the study of snow and

ice aerodynamic surface roughness during melting season at the August-one ice cap. We used both automatic digital photogrammetry and manual photogrammetry. Automatic methods allowed us to monitor daily variations in aerodynamic surface roughness; manual methods allowed us to characterize aerodynamic surface roughness variation along the main glacial flow line. We also recorded meteorological observations, so as to study the impact of weather conditions (e.g. snowfall or rainfall) on aerodynamic surface roughness. This data allowed a further effort to characterize variation of plot-scale $z_0$ from an energy balance perspective.

## 2. Data and methods

### 2.1 Study area and meteorological data

The August-one glacier ice cap is located in the middle of Qilian Mountains on the northeastern edge of the Tibetan Plateau (Figure 1a, 1b). The glacier is a flat-topped ice cap that is approximately 2.3 km long and 2.4 $km^2$ in area. It ranges in elevation from 4550 to 4820m a.s.l. (Guo et al., 2015). This study was conducted during the melting season of 2018, a season characterized by high precipitation. Energy balance analysis indicated that net radiation contribute 86% and turbulent heat fluxes contribute about 14% to the energy budget in the melting season. A sustained period of positive turbulent latent flux exists on the August-one ice cap in August, causing faster melt rate in this period (Qing et al., 2018).

Researchers had access to meteorological data that had been recorded continuously since September 2015, when an automatic weather station (AWS) was sited at the top of the ice cap (Table 1). The AWS measures air temperature, relative humidity, and wind speed at 2 and 4 m above the surface. Air pressure, incoming and reflected solar radiation, incoming and outgoing long wave radiation, glacial surface temperature (using an infrared thermometer) are measured at 2 m height. Mass balance is measured by a Campbell Scientific ultrasonic depth gauge (UDG) close to the AWS. An all-weather precipitation gauge adjacent to the AWS measures solid and liquid precipitation. All sensors sample data every 15 seconds. Half-hourly means are stored on a data logger (CR1000, Campbell, USA). Throughout the entire melting season (from June to September) researchers periodically checked the AWS station, to make sure that it remained horizontal and in good working order. During the entire study period, precipitation total was 261.3mm as measured at the AWS. Of that, 172.1 mm was snow or sleet and 89.2 mm was rainfall (Figure. 7 a).

### 2.2 Automatic photogrammetry

The study began with the placement of an automatic close range photogrammetry measurement apparatus in the middle of the ice cap (4700m (98° 53.4′ E, 39° 1.1′ N. See Figure 1b and Figure2). It was placed near the existing meteorological station. This was done on July 10, 2018. A wooden frame, 1.5 m wide, and 2 m long, was put on the ice surface. This frame served as a geo-reference control field (Figure. 3a). Four feature points demarcated the control field; three additional points served as check points. A Canon EOS 1300D cameras with an image size of 5184×3456 pixels was connected to the frame. The camera lens was set in wide-angle mode (focal length of 27mm). The f-stop was fixed at f 25 with an exposure time of 1/320s. The camera was programmed to automatically take seven pictures over a period of ten minutes. The photography was repeated at three-hour intervals from 9:00 AM to 18:00 AM, Beijing time. During the ten-minute photography periods, the camera moved

along a 1.5 m long slider rail. The camera was 1.7m above ice surface and moved along the control frame. The seven pictures taken during this period were merged to produce a picture of ice surface topography at millimeter scale (Figure 3b). This apparatus took pictures over a period of three months (July 12 to September 15, the melting season). Sixty-four days of data were recorded. Each daily photography series produced four sets of pictures (twelve hours, three hour intervals). The best-exposed photo sets were manually selected and used as that day's data. We also set up instrumentation to record incoming and reflected solar radiation. Samples were taken every 15 seconds; 10-minute means were stored on a data logger (CR800, Campbell, USA) located at a height of 1.5m. Surface elevation changes caused by accumulation and ablation was measured by a digital infrared hunting-video camera, which took pictures of ice-surface gauge stakes located near the automatic photogrammetry site.

## 2.3 Manual photogrammetry

Manual close-range photogrammetry was used to survey glacier surfaces at several different locations of the ice cap. Observations were made on four days: July 12 and 25, and later on August 3 and 28. It should be noted that when the July measurements were performed, the ice cap surface was partially snow-covered.

Channels account for only a small portion of the glacier surface area. These surfaces show extreme variability of $z_0$ (Rippin et al., 2015; Smith et al., 2016). For that reason, we distributed the manual photogrammetry study sites over the glacier surface in such a way as to cover most surface types and topographic regions without including any channels (Figure 1b). We photographed a total of thirty-six sites over the four days of observation.

Study plots were demarcated with a 1.1×1.1m portable square aluminum frame. Geo-reference of the point cloud was enabled using control points established by eight cross-shaped screws on the aluminum frame (Figure.3c). Photos (convergent photographs, low oblique photos in which camera axes converge toward one another) were taken at ~1.6 m distances, covering an area of ~1.75 m$^2$. Seven to twelve of such photos were taken at each survey site and surrounded the target area from different directions. The camera used was an EOS 6D 50mm, with fixed focal lens and an image size of 5472×3648 pixels. The f-stop was fixed at f 22 with an exposure time from 1/25 to 1/125 s.

## 2.4 Data processing

Structure-from-motion photogrammetry is revolutionizing the collection of detailed topographic data (Westoby et al., 2012; James et al., 2017). High resolution DEMs produced from photographs acquired with consumer cameras need careful handling (James and Robson, 2014). In this study, both manual and automatically derived photographs were imported into a software program, Agisoft Photoscan Professional 1.4.0. This software allowed us to estimate camera intrinsic parameters, camera positions, and scene geometry. Agisoft Photoscan Professional is a commercial package which implements all stages of photogrammetric processing (James et al., 2017). It has previously been used to generate three-dimensional point clouds and digital elevation models of debris-covered glaciers (Miles et al., 2017; Quincey et al., 2017; Steiner et al., 2018), ice surfaces and braided meltwater rivers (Javernick et al., 2014; Smith et al., 2016). In our study, we found that after new snowfall, it was

difficult to match feature points in the photo sets. Three days of automatic data could not be processed. We estimated $z_0$ data
for the missing days based on data from snowfall days at the automatic site.

**2.5 Aerodynamic roughness estimation**

Methods for measuring roughness at plot-scale were first developed by soil scientists (Dong et al., 1992; Smith, 2014). Metrics
such as the random roughness (RR) or root mean square height deviation ($\sigma$), the sum of the absolute slopes ($\Sigma S$), the
microrelief index (MI), and the peak frequency (the number of elevation peaks per unit transect length) were used. Later these
roughness indices were used to describe snow or ice surface roughness (Rees and Arnold, 2006; Fassnacht et al., 2009b; Irvine-
Fynn et al., 2014).

Current photogrammetry methods produce high-resolution three-dimensional topographic data. Earlier two-dimensional
profile-based methods for estimating surface roughness discard much of the potentially useful three-dimensional topographic
data (Passalacqua et al., 2015). Smith et al. (2016) were able to use equation (1), developed by Lettau (1969), to make better
use of the topographic data, using multiple point clouds and digital elevation models (DEM). Fitzpatrick et al. (2019) also
developed two methods for the remote estimation of $z_0$ by utilizing lidar-derived DEM.

In this method, $z_0$ is quantified as:

$$z_0 = 0.5 h^* \frac{s}{S} \qquad (1)$$

where: $h^*$ represents the effective obstacle height (m) and is calculated as the average vertical extent of micro-topographic
variations; $s$ is the silhouette area facing upwind (m$^2$); $S$ is the unit ground area occupied by micro-topographic obstacles (m$^2$);
and 0.5 is an averaged drag coefficient.

Based on the work of Lettau (1969), Munro (1989) simplified the equation (1) by assuming that h* can equal twice the standard
deviation of elevations in the de-trended profile, with the profile's mean elevation set to 0 meter. The aerodynamic roughness
length for a given profile then becomes

$$z_0 = \frac{f}{X}(\sigma_d)^2 \qquad (2)$$

Where $f$ is the number of up-crossings above the mean elevation in profile; X is the length (m) of profile, and $\sigma_d$ is the standard
derivation of elevations of profile.For manual photogrammetry, we put the aluminum frame horizontally over the ice surface,
the plot is detrended by setting the control points at z axis of the same values. For automatic photogrammetry, the control field
of wooden frame was also laid horizontally over the ice surface that lowered as the ice melted and maintained a horizontal
position between the control field and ice surface. A DEM based approach enables the roughness frontal area $s$ to be calculated
directly for each cardinal wind direction (Smith et al., 2016). The combined roughness frontal area was calculated across the
plot, the ground area occupied by micro-topographic obstacles is 1m$^2$. We used a DEM-based average ($\bar{z}_{0\_DEM}$) of four
cardinal wind directions to represent overall aerodynamic surface roughness. Based on the half-hour wind direction data at the

August-one ice cap, the daily upward wind direction DEM-based $z_{0\_DEM}$ was also estimated at the automatic photogrammetry site. Considering that wind direction changed during the day, in this case we selected the prevailing wind direction to calculate frontal area $s$. The prevailing upwind direction DEM-based $z_{0\_DEM}$ was applied to calculate turbulent heat flux. Using the Munro (1989) method, $z_{0\_Profile}$ was calculated for every profile (n=1000) in both orthogonal directions for each plot at the automatic photogrammetry site.

## 2.6 Snow and ice surface energy balance calculation

The temporal variation of $z_0$ at the automatic site was studied from energy balance perspective. The surface heat balance of a melting glacier is given by:

$$Q_M = Q_{is} - Q_{os} + Q_L + Q_E + Q_H + Q_P + Q_G \qquad (3)$$

Where, $Q_M$ is the heat flux of melting; $Q_{is}$ is the incoming shortwave radiation; $Q_{os}$ is the outgoing shortwave radiation; $Q_L$ is the net longwave radiation; $Q_E$ is the latent heat flux; $Q_H$ is the sensible heat flux; $Q_P$ is the heat from rain; and $Q_G$ is subsurface heat flux.

In a horizontally homogeneous and steady surface state, the surface heat fluxes $Q_E$ and $Q_H$ can be calculated using either the bulk aerodynamic approach or profile method, based on the Monin-Obukhov similarity theory (e.g., ; Arck and Scherer, 2002; Garratt, 1992; Oke, 1987). In this study, half-hour observations at 4 m level and daily upward wind direction DEM-based $z_0$ were used to calculate $Q_E$ and $Q_H$ based on the bulk method. The heat from rain is given by Konya and Matsumoto (2010):

$$Q_P = \rho_w\, C_W T_W P_r \quad (4)$$

Where, $\rho_w$ is the density of water(1000 kg m$^{-3}$); $C_W$ is the specific heat of water (4187.6 J kg$^{-1}$ K$^{-1}$); $T_W$ is the wet-bulb temperature (K); and $P_r$ is the rainfall intensity (mm). The subsurface heat flux $Q_G$ is estimated from the from the temperature-depth profile and is given by $Q_G = -k_T \frac{\partial t'}{\partial z'}$ where $k_T$ is the thermal conductivity, 0.4Wm$^{-1}$K$^{-1}$ for old snow and 2.2W m$^{-1}$K$^{-1}$ for pure ice (Oke, 1987).

In order to calculate $P_r$, we used the air temperatures recorded at the AWS. There is an elevation difference between the study site (4700 m) and the AWS (4790m); recorded air temperatures were corrected to account for the elevation difference, a lapse rate of -5.6 °C Km$^{-1}$ was applied based on observation nearby (Chen et al., 2014) . The ice cap is flat and open terrain so in this case wind speed and relative humidity at the study sites were assumed to be close to those observed at the AWS.

## 3. Results

### 3.1 Photogrammetry precision

We used seventeen plots to analyze the horizontal and vertical accuracy of our automatic photogrammetry, and thirty-one plots for our manual photogrammetry. Based on the Agisoft PhotoScan processing report, automatic photogrammetry average point density of the final plot point clouds was over 1,000,000 points m$^{-2}$. DEMs of 1mm resolution were generated at plot scale.

The average geo-reference errors fluctuated at around 1 millimeter (see Tables 2 and 3). Total RMSE of the automatic control points was $3.0\pm2.1$ mm, for check points $3.62\pm1.6$ mm. Vertical error for control points was $3.58mm\pm3.01mm$, and $4.83\pm2.9mm$ for check points (Tables 2 and 3). Standard deviation of control and check point errors are all within 15 mm (Figure 4a, 4c, 4e). Manual measurements average point density of the final plot point clouds was >6,000,000 points m$^{-2}$. DEM of 1 mm resolution was generated at plot scale. Root mean square error (RMSE) of 4 control points is $1.78\pm1.3$ mm (Table 1). Control points vertical accuracy of manual photogrammetry is about $1.65\pm1.3$ mm. Total RMSE of manual photogrammetry check points is $0.99\pm0.3$ mm, vertical accuracy is $0.66\pm0.3mm$ (see Tables 2 and3). Standard deviation for x, y and z axis were all within 5mm (Figure 4 b, 4d, 4f).

Note that the control and check point errors are larger for the automatic measurements than for the manual ones (See Figures 4). We believe that this is the case because, rather than using static f-stop and exposure times (as in automatic photogrammetry) researchers engaged in manual photogrammetry could adjust exposure time based on ice surface conditions. This allowed production of better quality photos even on cloudy or foggy days. The difference of survey design also caused more precise results for manual than automatic photogrammetry. For the automatic measurements, the camera was moving linearly, and the density of tie-points was much higher in the foreground compared to the background. For the manual method, photos were taken by surrounding the target area. This type of surface provided a much more robust elevation model and points density.

## 3.2 Aerodynamic surface roughness as measured by automatic photogrammetry

Data for ice surface roughness was collected by the automatic photogrammetry camera site from July 12 to September 15, a period covered the whole melting season. Profile and DEM data show that $z_0$ estimates vary by two orders of magnitude over the study period (Figure 5). The upwind DEM-based data showed a $z_{0\_DEM}$ varying from 0.1 mm to 1.99mm (mean: 0.55 mm). The average of four cardinal wind directions DEM data shows a $\bar{z}_{0\_DEM}$ varying from 0.1mm to 2.55 mm (mean: 0.57 mm). The average Munro profile based $z_{0\_Profile}$ varied from 0.03mm to 2.74 mm (mean 0.46 mm).

At the start of the observation period of July 12, snow covered the study site. As the snow melted, the ice cap surface $z_0$ increased. During this periods, $z_0$ dropped to around 0.1mm due to intermittent snowfall. On July 21, cryoconites appeared on patches of snow-crust, which led to patchy melt. From July 21 to 24, overall $\bar{z}_{0\_DEM}$ increased from 0.1mm to 1.6mm. By July 29, snow had disappeared from the study site; $z_0$ fluctuated but trended lower. From July 29 to August 5 bare ice covered whole field of view; $\bar{z}_{0\_DEM}$ ranged from 0.18 to 0.56mm. From August 6 to September 3 there was intermittent snowfall followed by melting; $\bar{z}_{0\_DEM}$ ranged from 0.1 to 1.0mm. From September 4 to September 14 $\bar{z}_{0\_DEM}$ showed an overall increase, reaching a maximum of 2.55 mm on September 8. There was intermittent snowfall during this period, which temporarily reduced $\bar{z}_{0\_DEM}$. $\bar{z}_{0\_DEM}$ which then increased thanks to patchy micro-scale melting. After September 14, snow covered the whole surface of the glacier and there was no melting and little fluctuation in $z_0$.

It should be clear that either $z_{0\_Profile}$ or $z_{0\_DEM}$ and $\bar{z}_{0\_DEM}$ varied following the same pattern during the melting season.. There were two peaks in $z_0$, both of which occurred in period of transition: snow surface turning to ice around July 24 and ice surface turning to snow on September 8. On July 24 and again on September 8 and 13, glacier surfaces featured cryoconite holes and snow crust. Both the automatic and manual observations showed the same pattern: maximum $z_0$ at snow-ice transition belt during partially snow-covered periods.

### 3.3 Surface roughness as measured by manual photogrammetry

No wind direction measurements were carried out during manual photogrammetry. In this case, we presented an average of four cardinal directions to represent ice aerodynamic surface roughness. Analysis indicated that $\bar{z}_{0\_DEM}$ proved to have an interesting relationship with altitude. $\bar{z}_{0\_DEM}$ was highest in the transition zone between snow cover and ice. This zone moved up the ice cap during the melting season. On July 12, ice surface roughness decreased from 3.2mm to 0.25mm as altitude increased (Figure. 6a, r= 0.8429, P=0.0006<0.01). Near the ice cap terminals of 4590m, the ice surface featured porous

snow/ice and many cryoconite holes. As altitude increased, the number of cryoconite holes decreased and snow coverage increased. At 4700m the ice surface was predominantly snow covered, and only a few small patches were bare of snow. On July 25, ice surface roughness fluctuated between 0.27 to 0.65 mm at the ice cap terminals (4593m). At ~4700m, roughness increased to 1.85mm. Above that point, roughness gradually decreased to 0.25mm at the ice cap top, which was covered by snow (Figure 6b).

On August 3, the August-one ice cap was predominantly bare ice; there was scattered snow crust at the ice cap top. The ice surface, (terminals to top) showed a heavy deposit of cryoconite. Potogrammetric data collected manually revealed that ice surface roughness increased with altitude (Figure. 6c, r=0.7). From terminals to top, $z_0$ varied from 0.06 mm to 2.2 mm. On August 29, the ice cap surface roughness showed no significant correlation with altitude (Figure. 6d, r=-0.03). $\bar{z}_{0\_DEM}$ varied from 0.2 mm to 0.98 mm (Figure6 d). When we compare the results of the four surveys, we see that ice surface roughness was

variable. Maximum $z_0$ was seen at the snow and ice transition zone, where the ice surface featured both cryoconite holes and clean snow crust. Snow crust would have inhibited melting; cryoconite would have increased it. It is thus understandable that surface roughness would have been greater in such an area. Bare ice or snow cover both result in comparatively less roughness.

### 3.4 $Z_0$ and weather

Figure 7 compared $\bar{z}_{0\_DEM}$ and corresponding meteorological conditions of precipitation, air temperature, downward solar

radiation, relative humidity and wind speed. Detailed analysis indicates snowfall was recorded from July 12 to 24. In general, snowfall reduced roughness if it resulted in a fully snow-covered surface. However, if a patchy, shallow snow cover was formed, it tended to increase $z_0$ after a short drop. For example, on August 11 and 12, two successive sleety days created a patchy snow cover which soon increased $z_0$. Between July 26 and August 31 there were sixteen rainfall events, which tended to lower ice surface $z_0$.

Daily temperatures during the study period ranged from -6.5 °C to 7.1 °C (mean: 1.3, Figure. 7c). It was 1.2 °C on July 11. It increased to 3.6 °C on July 24 (the date when $z_0$ was highest). It continued increasing until July 29, when it reached its highest annual of 7.1 °C. During this period $z_0$ continuously declined. From July 28 to end of August temperatures fluctuated between -0.3 to 5.7 °C with no evident trend. $\bar{z}_{0\_DEM}$ also fluctuated slightly, showing no obvious trend. In September air temperature quickly dropped from 0.6 to -6.5 °C. There were large fluctuations in $z_0$ during this period. The largest fluctuations appeared

when air temperatures dropped from positive to negative.

Daily downward mean solar radiation fluctuated dramatically during the study period due to cloud and overcast (Figure. 7d). Incident solar radiation fluctuated between 129W m$^{-2}$ and 753 W m$^{-2}$ (mean: 469 W m$^{-2}$). From July 29 to end of August, the weather was cloudy, warm, calm, and humid most of the time (Figure. 7b, 7c, 7e 7f), and $\bar{z}_{0\_DEM}$ was relatively stable except

when there was intermittent snowfall-induced fluctuation. After in September, the weather was again becoming cold and dry and $z_0$ was quite variable.

### 3.5 Ice-surface energy balance at automatic $z_0$ observation study site

The following section analyzes the changes in surface energy balance at the automatic site. Meteorological observation records allowed us to study the factors that control ice surface roughness. Net radiation varied from -9.7 to 260.2 W m$^{-2}$ (mean: 95.3 W m$^{-2}$) during the study period. This constituted the largest energy flux affecting glacier-surface energy balance. It accounted for 84% of total incoming flux (Figure. 8). Net radiation was relatively low in the first thirteen days of the study period (mean: 69.3 Wm$^{-2}$), when the glacier surface was covered with snow. In the succeeding five days, net radiation increased to 103.9 W m$^{-2}$. At this time the ice surface exhibited a patchwork of snow, ice, and cryoconite. From July 29 to August 5 the surface of the study site was composed of ice with a dusting of cryoconite. Net radiation reached a height of 183 Wm$^{-2}$. There was intermittent snowfall from August 6 to September 8. Net radiation dropped to a mean 93 Wm$^{-2}$. Snow cover then appeared and net radiation dropped to a low of 46 Wm$^{-2}$.

Bulk method estimated results indicate that sensible heat ($Q_H$) was the second largest energy-flux component of in surface energy balance during the study period (Figure 8). The sensible heat daily mean varied from -7.1 to 66.3 W m$^{-2}$. It accounted for -28% to 32% (mean: 15%) of the net energy flux. Latent heat was generally small throughout the study period. Daily mean of latent heat varied from -80.1 to 11.1 W m$^{-2}$ (mean: -13.2 W m$^{-2}$). It account for a mere 0.9% for the total incoming flux. It was negative from July 11 to 26 when the ice surface was snow covered. After July 26 the latent heat was mainly positive in the following ten days (ice surface was pure ice or partially snow covered). From August 6 to the end of the study period (September 15) it was predominantly negative.

From July 25 to August 5 rainfall energy varied from 0 to 11.7 W m$^{-2}$ (mean: 0.3W m$^{-2}$). Rainfall accounted for a mere 0.2% of total incoming flux. One event accounted for much of the total: on July 28 a 31mm rainfall event added a flux of 11.7 W m$^{-2}$, which resulted in visible smoothing of the ice surface (Figure 9). Compared to other energy components, $Q_G$ was very small, with a daily mean of -0.65 W m$^{-2}$ and a maximum and minimum of -0.4 and -2.1 W m$^{-2}$, respectively.

### 3.5.5 Surface ablation modeled versus observed

Based on the previously listed measurements of energy fluxes we calculated the probable surface ablation at the automatic photogrammetry site. We took into account observed net radiation, bulk method calculated turbulent heat fluxes, heat from rainfall, and subsurface heat flux. There was good agreement between the model and observed results (Figure 10).

Figure 11 shows the relationship between estimated daily upward wind direction DEM-based $z_{0\_DEM}$ and the main energy flows. Scatter diagrams showed a positive relationship between $z_{0\_DEM}$ and net shortwave radiation (Figure 11a, r=0.1) and a significant negative relationship between $z_{0\_DEM}$ and net longwave radiation (Figure 11b, r=-0.35), Graphing $z_{0\_DEM}$ vs. bulk method estimated latent heat showed a significant negative exponential relationship (Figure 11d, r= -0.35). The scatter diagram showed no significant relationship between $z_{0\_DEM}$ and the bulk method estimated sensible heat (Figure 11c). The average of the Munro profile based $z_{0\_profile}$ and DEM based $\bar{z}_{0\_DEM}$ and the main energy items are also analyzed respectively. Scatter

diagrams showed significant negative relationship between $z_{0\_profile}$ and net longwave radiation (Figure1s b, r=-0.5). Graphing $z_{0\_profile}$ vs. the bulk method estimated sensible heat showed a significant negative exponential relationship (Figure 1s d, r=-0.69). These scatter diagrams showed no significant relationship between $z_{0\_Profile}$ and the bulk method estimated sensible heat (Figure 11c, 11e). $\bar{z}_{0\_DEM}$ vs. the bulk method estimated latent heat showed a significant negative exponential relationship (Figure 2s d, r= -0.44). The scatter diagrams between $\bar{z}_{0\_DEM}$ and net shortwave radiation, the bulk method estimated sensible heat showed no significant relationship.

Because net shortwave radiation and turbulent heat fluxes were the main energy fluxes affecting ice surface roughness, we calculated a turbulent heat proportion index:

$$L_S=(Q_H+Q_E+Q_P)/(Q_{is}-Q_{os}) \quad (5)$$

Note that aerodynamic surface roughness on days when snow fell was strongly affected by the amount of the snowfall. If we exclude snowfall days and snow covered period, we see a significant exponential relationship between ice surface $z_{0\_DEM}$ and $L_S$ (Figure 12a, r= -0.34). Scatter diagrams showed significant exponential relationship between ice surface $z_{0\_Profile}$ and $L_S$ and net longwave radiation (Figure12b, r=-0.69). $\bar{z}_{0\_DEM}$ vs. $L_S$ also showed a significant exponential relationship (Figure 12c, r=-0.46). Scatter diagrams in Figure 12 also showed $z_0$ did not keep decreasing when $L_S$ was above 0.2. $z_{0\_DEM}$, $z_{0\_Profile}$ and $\bar{z}_{0\_DEM}$ was around 0.56±0.21mm, 0.33±0.03 mm and 0.6±0.26 mm, respectively.

The $z_0$ ($z_{0\_DEM}$, $z_{0\_Profile}$ $\bar{z}_{0\_DEM}$) vs. $L_S$ graph indicates that when turbulence and rainfall heat increased, aerodynamic surface roughness decreased. As soon as $L_S$ is above 0.2, the ice surface will not keep smoothing and $z_0$ sustained its lowest stage. Time series correlation of all main energy items and $z_{0\_Profile}$ were performed. Table 4 shows an example of the lagged correlations between $z_{0\_profile}$ and five variables. The $z_0$ and net shortwave radiation displayed a positive correlation with 0 to 1 days lag time. The $z_0$ response to $Q_E$ with a correlation of -0.6 showed a lag of 0 to 1 days. The $z_{0\_Profile}$ also had a negative relationship with $Q_L$ with no lag or 1 day lag time. The $z_{0\_Profile}$ response to $L_S$ with a correlation of -0.58 was with a lag of 0 to 2 days. 0 to 2 days lag time gives an indication of the main energy items efforts limitations over ice surface $z_0$. In other words, a sunny and cold day facilitates rough ice surfaces; warm and cloudy days tend to produce a smoother ice surface. When net shortwave radiation is higher, and if latent and sensible heat were smaller, $z_0$ would tend to be higher for the next 2 days. When net shortwave radiation is smaller, as on cloudy days, any snowfall or rainfall is usually associated with smaller $z_0$ for the following 2 days. Under a negative $Q_M$, the surface $z_0$ would be not affected by melting process.

## 4. Discussion

### 4.1 Automatic and manual photogrammetric methods

Photogrammetric techniques such as Structure from Motion (SfM) (James and Robson, 2012) and Multi-view Stereo (MVS) represent a low-cost option for acquiring high-resolution topographic data. Such approaches require relatively little training and are extremely inexpensive (Westoby et al., 2012; Fonstad et al., 2013; Passalacqua et al., 2015). We used both automatic and manual photogrammetric methods to sample spatial and temporal $z_0$ variation at the August-one ice cap. Adjustments to exposure time based on ice surface conditions and survey design of the area surrounding the target made the manual

photogrammetry is more precise than automatic photogrammetry (Tables 2 and 3). However, precision is not always the major concern. The glacier surface was a harsh, even punishing environment for the researchers doing manual photogrammetry. In addition, manual photogrammetry took much longer. Automatic methods reduced hours of field work, spared researchers, and produced nearly continuous data. Cloudy or frosty weather affected automatic photogrammetry exposures, and heavy snowfalls resulted in a texture-less surface. Nevertheless, it is likely that photogrammetry techniques will continue to improve and that these drawbacks may be mitigated.

**4.2 Spatial and temporal variability of $z_0$**

Previous studies of glacier surfaces roughness have rarely covered the whole glacier, from terminals to top, in one melting season (Föhn, 1973; Smeets et al., 1999; Denby and Smeets, 2000; Greuell and Smeets, 2001; Albert and Hawley, 2002; Brock et al., 2006; Smeets and Van den Broeke, 2008; Smith et al., 2016). This whole-glacier study allowed us to follow the movement of the transition zone, where snow was melting and exposing ice, from terminals to top. The transition zone moved up as the melting season proceeded, so roughening the surface of the glacier and raising $z_0$.At the start of the melting season, snow cover first disappeared, leaving an ice surface, at the terminals end of the August-one ice cap, that is, at the lower altitude. This newly exposed surface was rougher ($z_0$ was higher) than on the upper part of glacier, which was still snow covered (see the black line Figure 6a for $z_0$ distribution at different altitudes). As the snowline shifted to higher altitudes, ice surface increased, as did $z_0$ (see the dashed black curve in Figure 6b). As the melting continued, the snow and ice transition belt reached the top of glacier (see the dotted curve in Figure 6c). When the ice cap was completely free of snow, $z_0$ and elevation were no longer correlated (see the dotted-dashed line in Figure 6d). In summary, maximum $z_0$ was recorded at the cross-glacier transition zone between snow and ice. This zone shifted from lower altitude to higher altitude, from terminals to top, during the melting season. The spatial pattern of $z_0$ distribution affected turbulent fluxes. The transition zone had maximum $z_0$ and the zone also migrated across much of the glacier, highlighting the importance of transient surface characteristics.

Micro-topography, wind profile, and eddy covariance methods generate a wide range of $z_0$ values for snow and ice surfaces (Grainger and Lister, 1966; Munro, 1989; Bintanja and Broeke et al., 1995; Schneider, 1999; Hock and Holmgren, 2005; Brock et al., 2006; Andreas et al., 2010; Gromke et al., 2011)., In this study, $z_{0\_profile}$, $z_{0\_DEM}$, and $\bar{z}_{0\_DEM}$ showed similar variation pattern during the melting season. The difference of $z_{0\_profile}$, $z_{0\_DEM}$, and $\bar{z}_{0\_DEM}$ were within one order of magnitude. The latent and sensible heat calculated by $z_{0\_profile}$, $z_{0\_DEM}$, and $\bar{z}_{0\_DEM}$ were highly relevant among these methods. The automatic photogrammetry estimated $z_0$ for snow-covered surfaces ranged from 0.1 to 0.55. New snowfall at snow surface in July formed the lowest $z_0$ values. Previous studies have shown that freshly fallen snow is subject to rapid destructive metamorphism (McClung and Schaerer, 2006), which can dramatically change the roughness of fresh snow surfaces (Fassnacht et al., 2009b). Our study showed that $z_0$ followed an increasing trend during melting season. Intermittent snowfall first decreased snow surface $z_0$, which then began to increase as the snow surface deteriorated. In the data from Clifton et al. (2008), snow surface $z_0$ was estimated at between 0.17 to 0.6 mm in a wind tunnel experiment. In an analysis of ultra-sonic anemometer recorder data over snow-covered sea-ice, Andreas et al. (2010) found $z_0$ values ranging from $10^{-2}$ to $10^{1}$ mm. In a wind-tunnel experiment of fresh snow with no-drift conditions, Gromke et al. (2011) estimated $z_0$ to be lying between 0.17 to 0.33 mm with no apparent dependency on the friction velocity. Our snow surface data showed $z_0$ values fluctuated between 0.03 to 0.55

mm, consistent with some of those wind-tunnel studies. The scatter of $z_0$ data reported in some studies is quite large, with a range of $10^{-2}$ to $10^1$ mm. The result may be attributed to the occurrence of snow drift, a transitional rough-flow regime and large uncertainties in the estimation of friction velocities that propagate to the computation of $z_0$ (Andreas et al., 2010; Gromke et al., 2011). On the contrary, the small scatter in our data was induced only by the natural variability of snow-surface roughness. For patchy snow-covered ice surfaces, $z_0$ varied from 0.5 to 2.6mm and ice surface $z_0$ varied from 0.24 to 1.1mm. During the melting season, there were no blowing snow events and snow surface $z_0$ was relatively smaller than in patchy snow-covered surface or ice surface. Ice surface $z_0$ was generally larger than snow surface and smaller than patch snow-covered surface. Our results match values reported in studies reporting results ranging from. 0.1mm to 6.9mm in Qilian mountain glaciers (Guo et al., 2018; Sun et al., 2018). Our results showed that $z_0$ reached its maximum at the end of the summer melt, which matched wind profile measurements by Smeets and Broeke (2008).

The aerodynamic surface roughness is influenced by both boundary layer and the surface. In this study, the microtopographic estimated aerodynamic surface roughness only considers surface topography at plot scale, but its variability influenced by its surrounding topography and boundary layer. Thus, the results of $z_0$ estimated in this study still need validated by wind tower or eddy covariance observations. However, microtopographic roughness metrics are a very strong proxy for $z_0$ (e.g. Nield et al, 2013), so we have much more confidence in the temporal and spatial variability presented by this work.

**4.3 Effects of surface energy balance components on aerodynamic surface roughness**

Aerodynamic roughness is associated with the geometry of ice roughness elements (Kuipers, 1957;Lettau, 1969;Munro, 1989). Surface geometry roughness develops due to local melt inhomogeneities in melting season. In early work, researchers argued that a variety of ablation forms, such as sun cups, penitents, cryoconite holes or dirt cones are formed by the sun (Matthes, 1934; Lliboutry, 1954; Mcintyre, 1984; Rhodes et al., 1987; Betterton, 2000). These ablation forms develop in regions with bright sunlight and cold, dry weather conditions are apparently required (Rhodes et al., 1987). These structures are observed to decay if the weather is cloudy or very windy (Matthes, 1934; Lliboutry, 1954; Mcintyre, 1984).

The August-one ice cap dust concentrations are high in the melting season. Cryoconites are unevenly distributed over the ice surface leading to differential absorption of shortwave radiation at microscale. This process results in the roughening of the ice surface; a process that enhances turbulent heat exchange across the atmospheric boundary layer-ice interface. When the air temperature is above 0 °C, the ice surface keeps melting. The turbulent heat smooths the ice surface and increases the cryoconite concentration over the ice surface and decreases ice surface albedo, enhancing shortwave radiation absorption (Figure 9). This roughening and smoothing process makes ice surface $z_0$ to fluctuate at around 0.56 mm as long as the air temperature is above 0 °C. When temperature drops below 0 °C, bright sunlight and dry weather shutdown the ice surface smoothing process. The shortwave radiation induces even rougher ice and larger $z_0$ until snow covers the ice surface. At the August-one ice cap, the turbulent heat contributes a small portion of incoming energy, but the smoothing ice surface process decreases ice surface albedo and seems enhance ice surface shortwave radiation. The $z_0$ fluctuation in the melt season is similar with cryoconite holes developing when the radiative flux is dominant and decaying when turbulent heat is dominant (McIntyre, 1984; Takeuchi et al., 2018). The glacier surface energy balance components vs. $z_0$ analysis in this study confirms that main energy items of net shortwave radiation and turbulent heat flux affect the same day and following 2 days $z_0$. This study found an exponential relationship between $z_0$ and $L_S$. The delicate role of $z_0$ played in the ice surface balance is still not fully known. Further

comparative studies are needed to investigate the $z_0$ variation through eddy covariance, profile method and DEM-based $z_0$ estimation.

## 5. Conclusions

Manual and automatic measurements of snow and ice surface roughness at the August-one ice cap showed spatial and temporal variation in $z_0$ over the melting season. Manual measurements, taken from terminals to glacier top, show that the nature of the surface cover features are correlated with $z_0$ rank in this order: transition region > pure ice area or pure snow area. The transition region forms a zone of maximum $z_0$, which shifts, over the melting season, from terminals to top. The observed $z_0$ vs energy items analysis indicated that $L_S$ (turbulent heat index) was also an important determinant of ice aerodynamic surface roughness. Aerodynamic surface roughness is a major parameter in calculations of glacier-surface turbulent heat fluxes. In previous studies investigators used a constant $z_0$ value for the whole surface of the glacier. This study captures much smaller scale variation spatial and temporal glacier surface aerodynamic roughness through automatic and manual photogrammetric observations. Such close observation of variation in $z_0$ certainly enhanced the accuracy of the surface energy balance models developed in the course of this study.

Of course, this study carried out at the ice cap with neat ordering of the annual layers. The August-once ice cap moved slowly and no crevasses were formed over the ice cap and channels were not considered in this study. In this case, a moderate variation of $z_0$ was estimated than it would be for debris covered glaciers (Miles et al., 2017; Quincey et al., 2017). Uneven or heterogeneous ice surface such as sastrugis, crevasses, channels, and penitents could greatly affect ice surface aerodynamic surface roughness and it would be hard to estimate its $z_0$ based on a profile method. SfM estimation of $z_0$ might be a good choice at macro-scale. In the accumulation season, more attention would be needed to be paid to spatial and temporal variations of $z_0$ as $z_0$ is a key parameter for sublimation calculation during this period. Studies have indicated that the Lettau (1969) approach calculated $z_0$ dependent on plot scale and resolution. In this study, we only select $1 \times 1$ m scale at 1mm resolution to study its spatial and temporal variability. Further comparative studies of $z_0$ are needed at different scales and resolutions.

*Data availability*. All of the observation and model input and output data presented in this study are available upon request to the corresponding author (Rensheng Chen, crs2008@lzb.ac.cn).

*Author contributions*. **JL and RC** designed the study and wrote the paper. **JL and CH** carried out field manual photogrammetry observations.

*Competing interests.* The authors declare that they have no conflict of interest.

*Acknowledgements.* This study was supported by the National Natural Sciences Foundation of China (41877163, 41671029). We thank the two reviewers for their insightful comments and ideas to improve the paper.

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

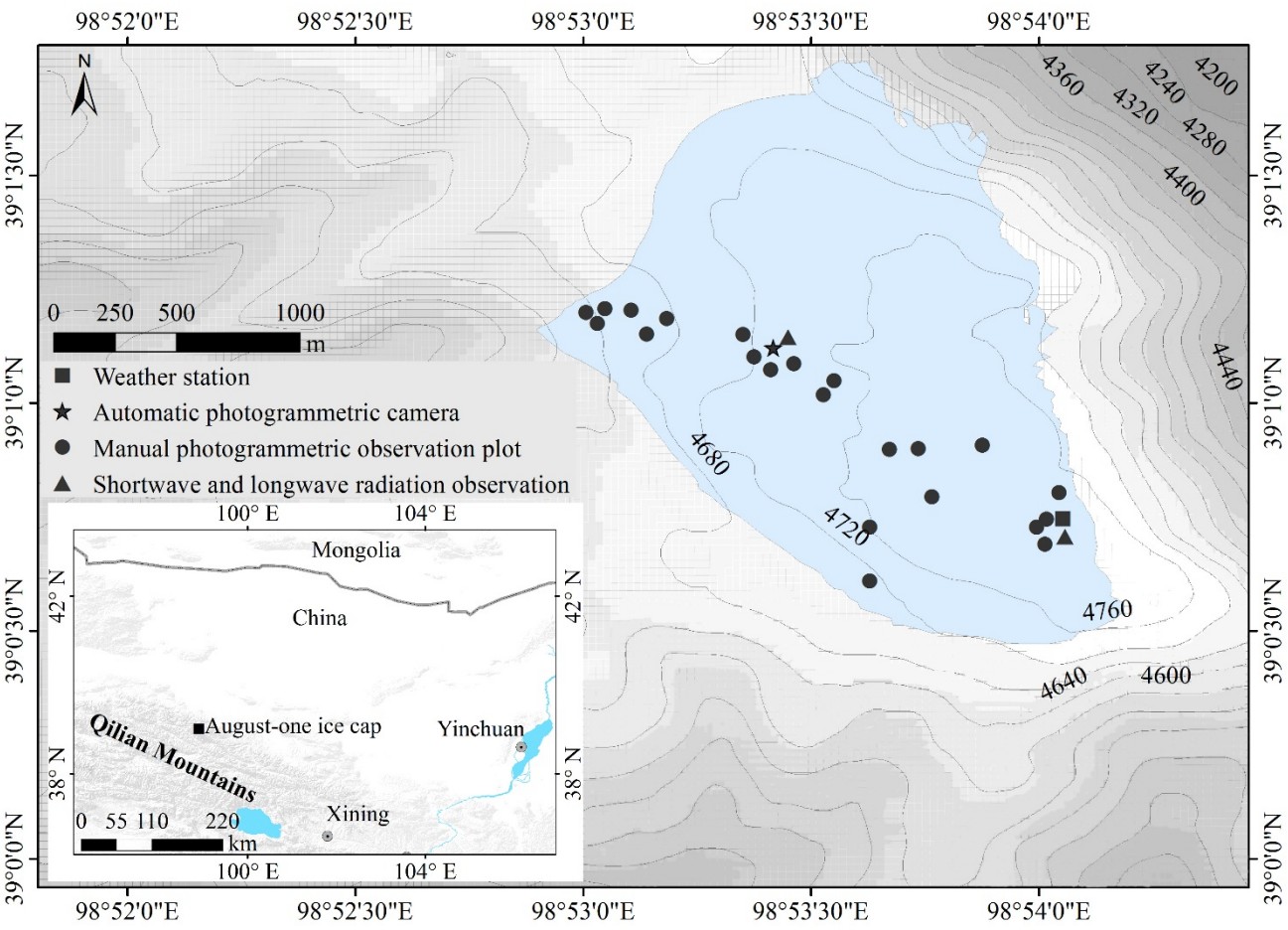

**Figure 1. Location of ice cap and study sites. (a) Location of the August-one glacier, (b) Locations of the AWS, automatic and manual photogrammetry plots, and shortwave observation platforms.**

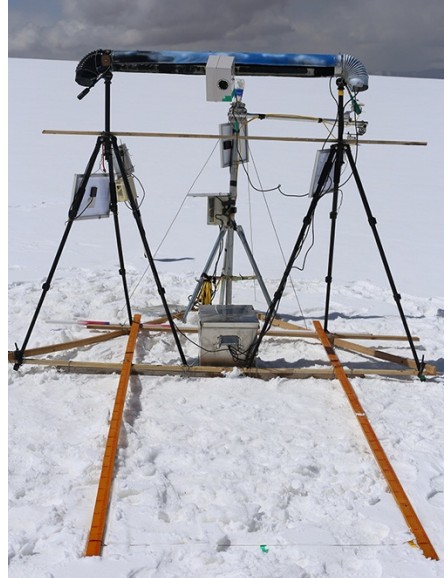

605        **Figure 2 The automatic photogrammetry device at the August-one ice cap.**

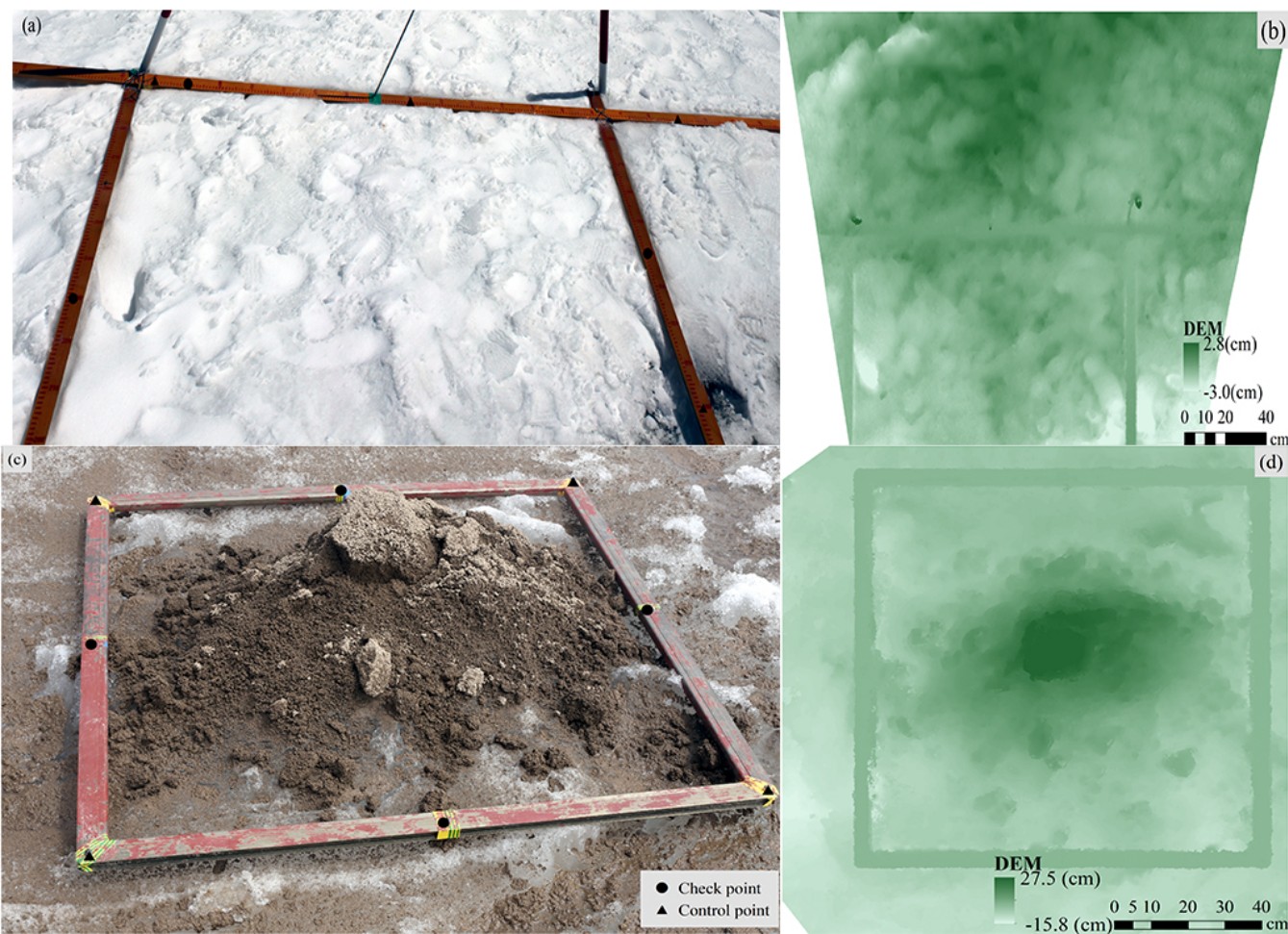

Figure 3. Frames used for automatic and manual photogrammetry. (a) Wooden frame in situ applied in automatic photogrammetry, four control points and three check points are shown on the frame; (b) Detrended DEM for the corresponding snow surface of Figure3a; (c) Manual observation plot, four control points and four check points shown on the aluminum frame. Ice surface hummocky was covered with cryoconites. (d) Detrended DEM for the corresponding cryoconites surface of Figure 3c.

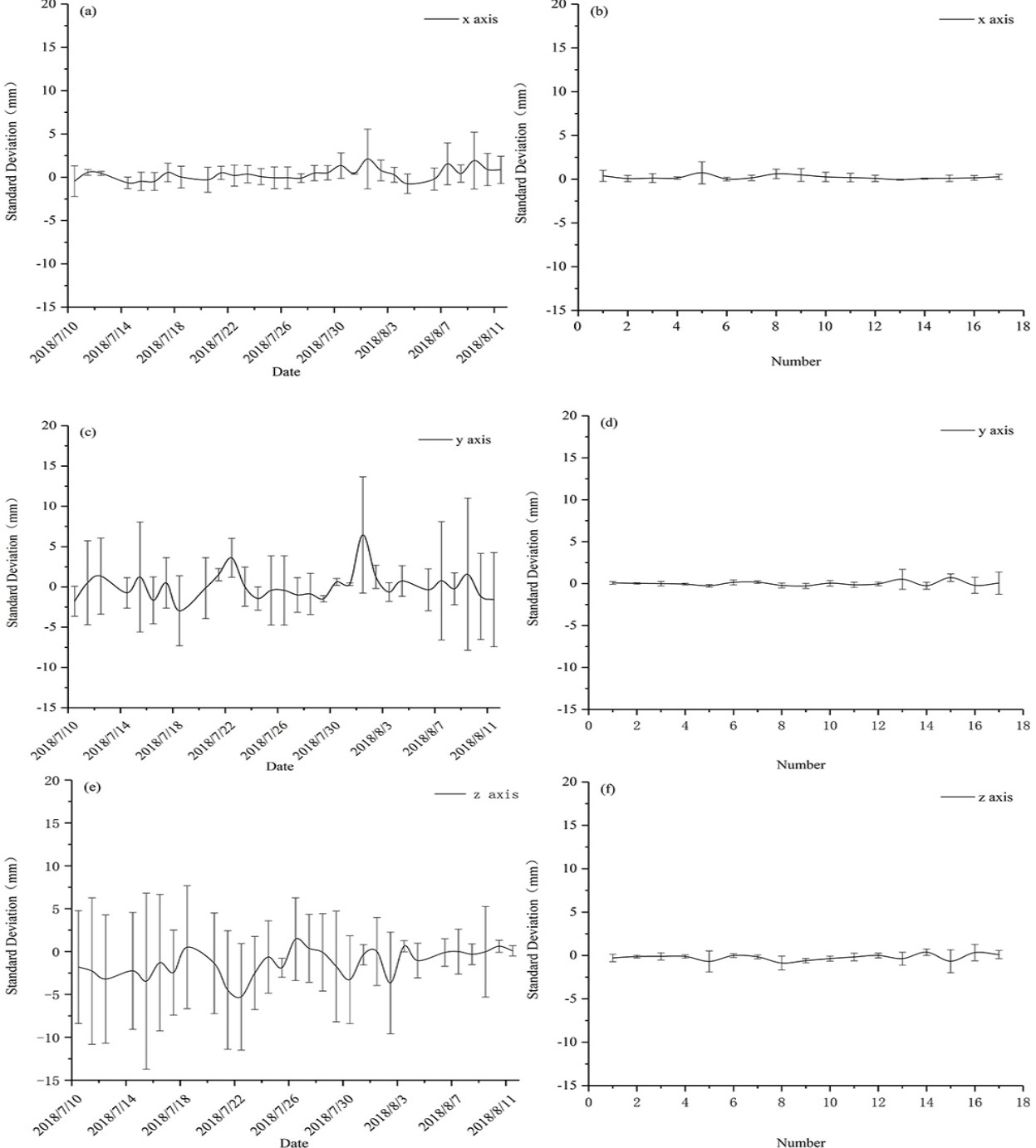

Figure 4. Automatic and manual photogrammetry checkpoint errors. (a), (c) and (e) are automatic photogrammetry standard deviation for x, y and z axis. (b), (d), and (f) are manual photogrammetry standard deviation for x, y and z axis.

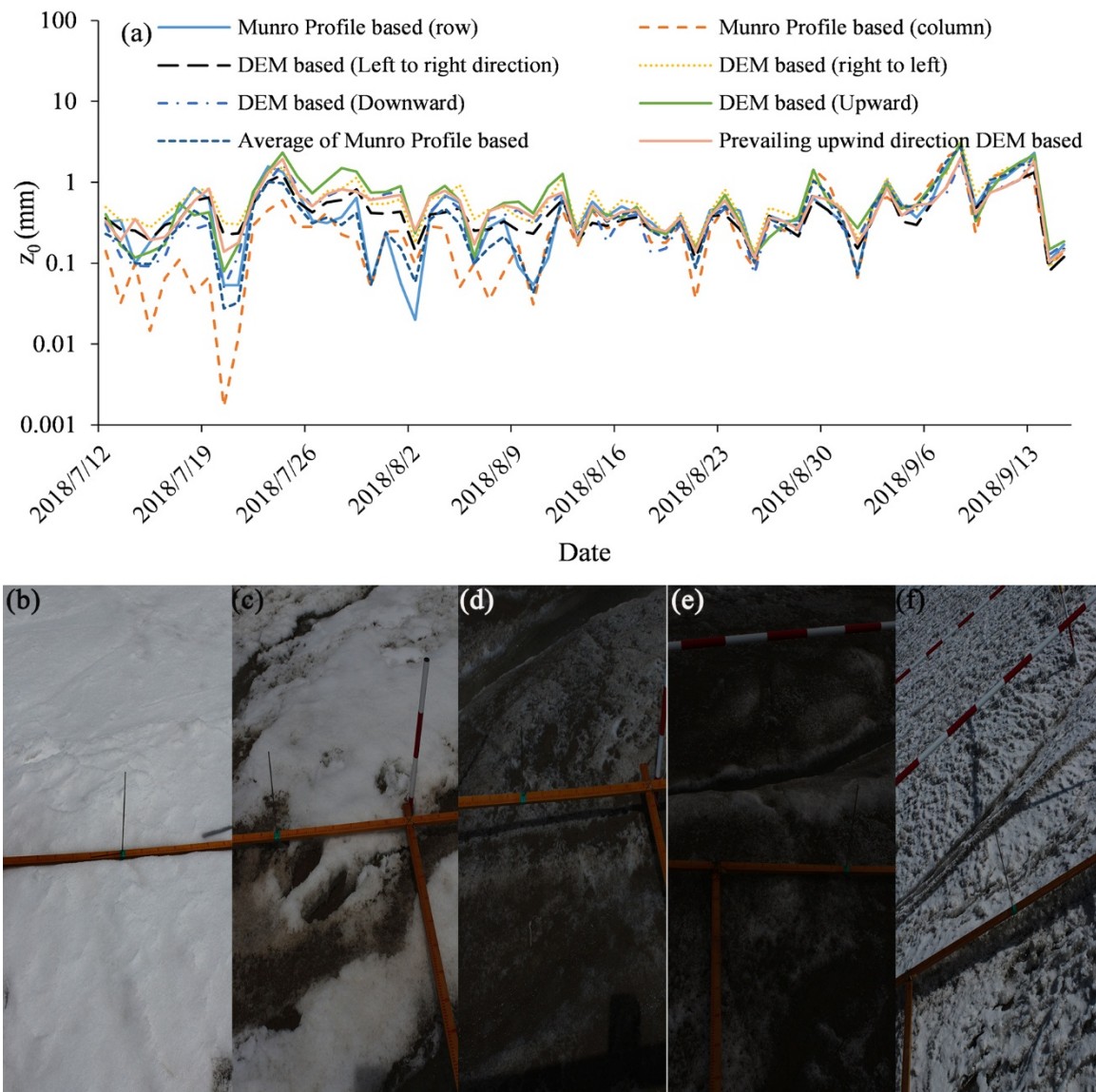

**Figure 5. (a) variation of glacier surface aerodynamic roughness over time at the automatic observation site for DEM based and Munro (1989) profile based approach; photo (b) showed snow covered surface on July 13, photo (c) showed partially snow covered surface on July 23 with cryoconite holes, (d) and (e) showed smooth ice surface on August 1 and August 30, (f) showed rough ice surface on September 13**

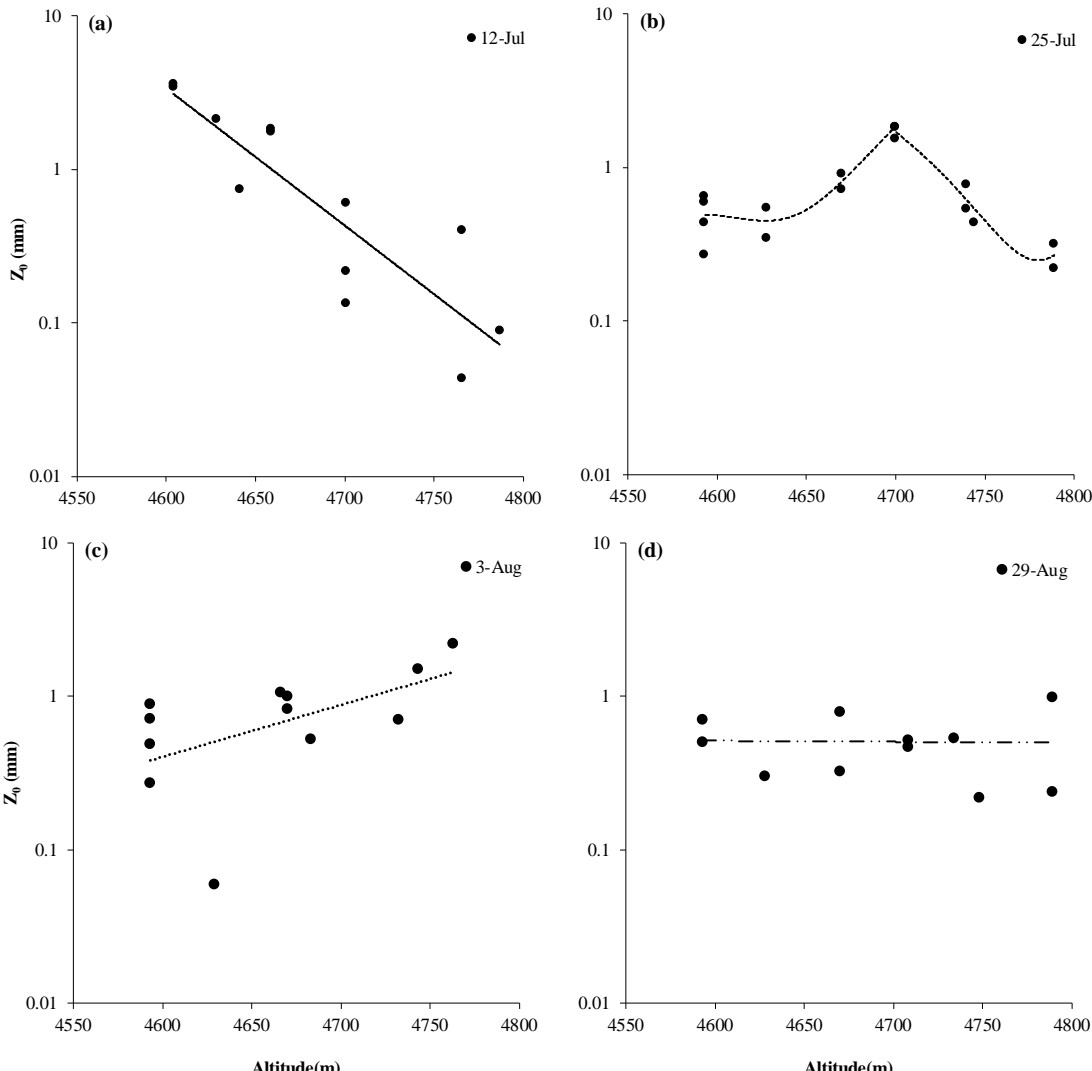

**Figure 6. Surface roughness vs. altitude, (a) As observed on 12 July, (b) As observed on 25 July, (c) As observed on 3 August, (d) As observed on 28 August.**

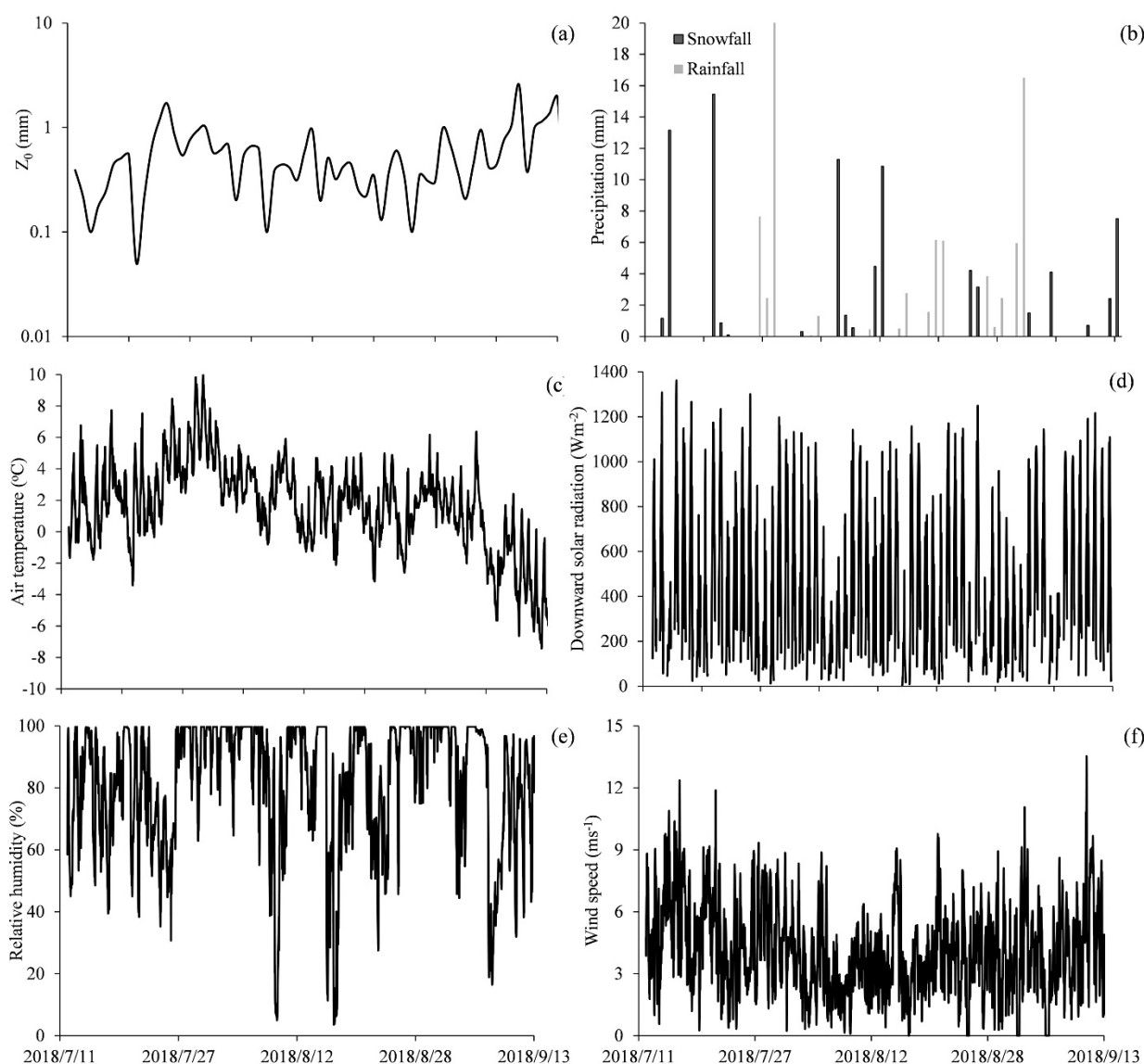

**Figure 7. Weather conditions at AWS over study period. (a) Precipitation, (b) Air temperature, (c) Incident solar radiation, (d) Relative humidity, (e) Wind speed.**

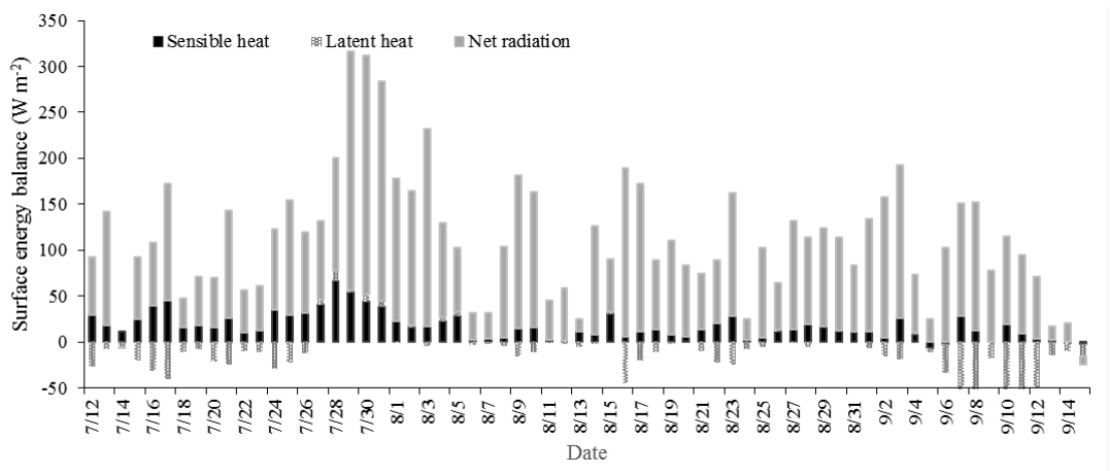

**Figure 8 Daily mean of energy balance at the middle of glacier study site close to the automatic photogrammetry site.**

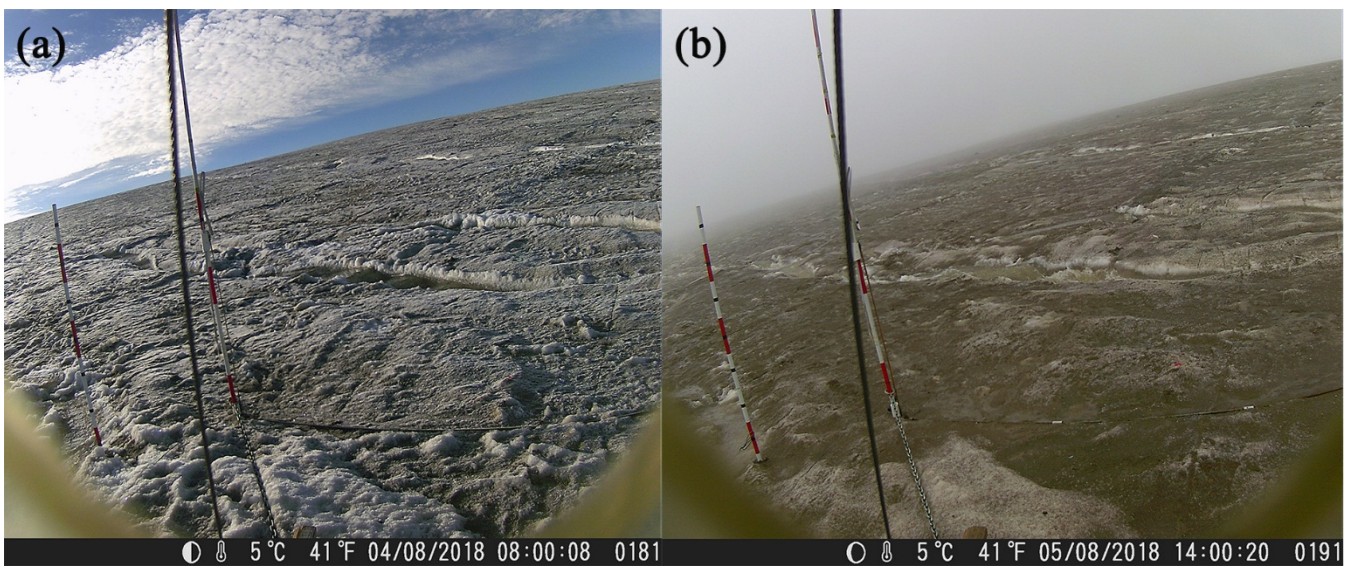

**Figure 9 Ice surface overview at the automatic photogrammetry site before and after a strong rainfall event captured by an automatic digital infrared hunting-video camera, (a) photograph before the rainfall event on August 4 of 2018, and (b) photograph after the strong rainfall event on August 5 of 2018**

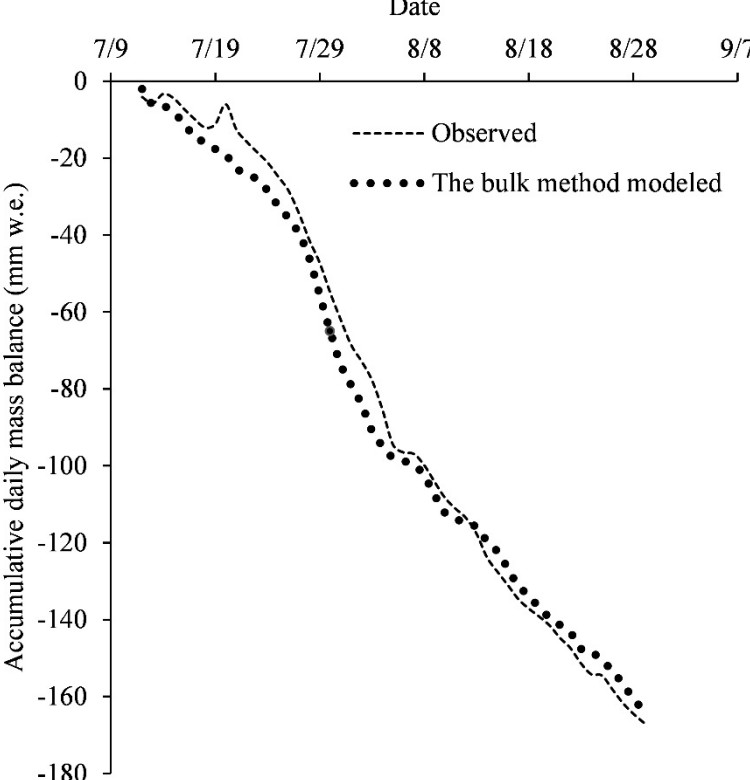

**Figure 10. Comparison of daily mass balance observed and daily mass balance as modeled. Mass balance measurements were taken from 12 July to August 29. Measurements of surface lowering were converted into water equivalents using density values.**

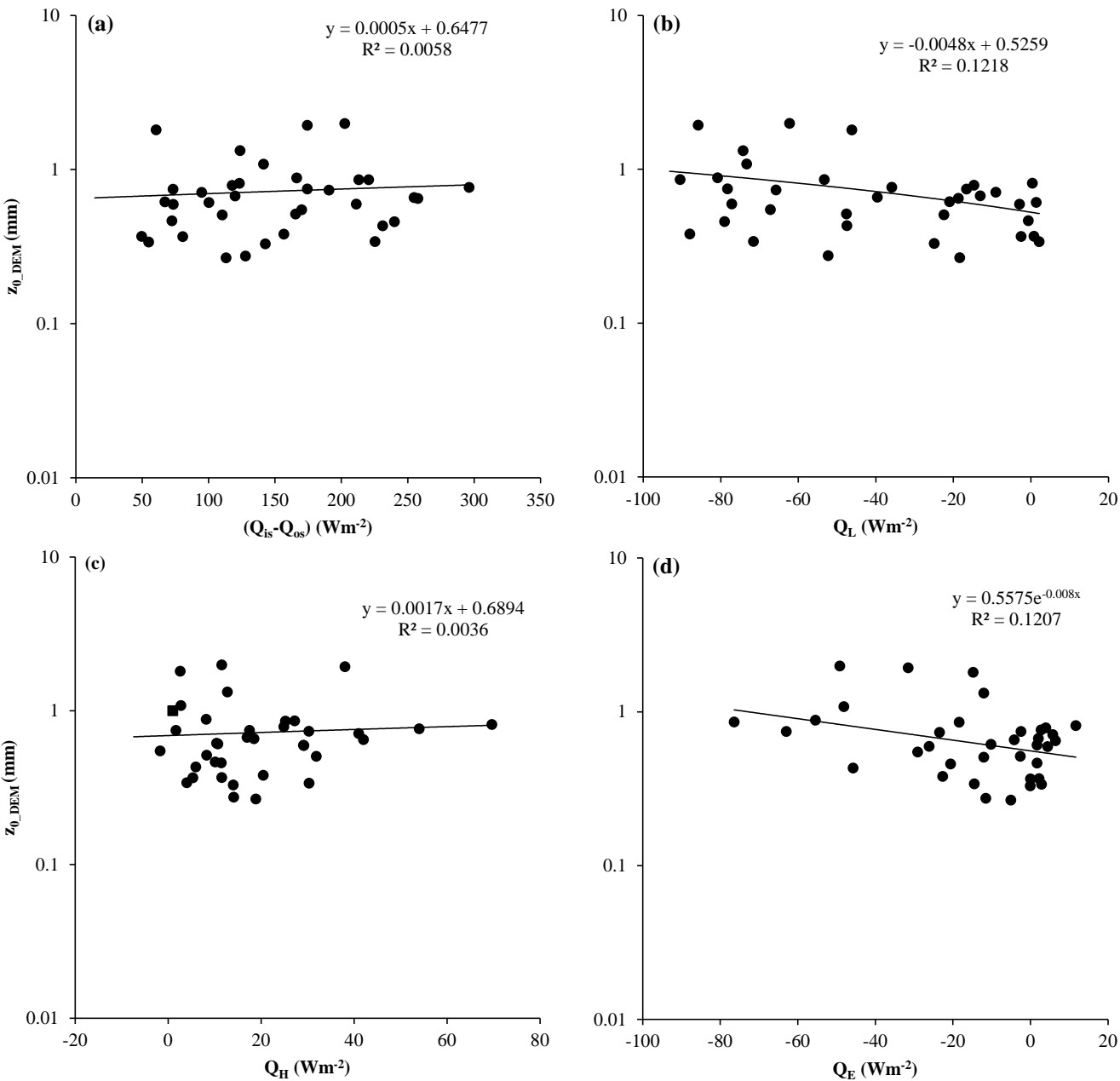


**Figure 11. Daily upward wind direction DEM-based $z_{0\_DEM}$ vs. energy inputs. (a) $z_{0\_DEM}$ vs. net shortwave radiation, (b) $z_{0\_DEM}$ vs. net longwave radiation, (c) $z_{0\_DEM}$ vs. the bulk method calculated sensible heat, (d) $z_{0\_DEM}$ vs. the bulk method calculated latent heat.**

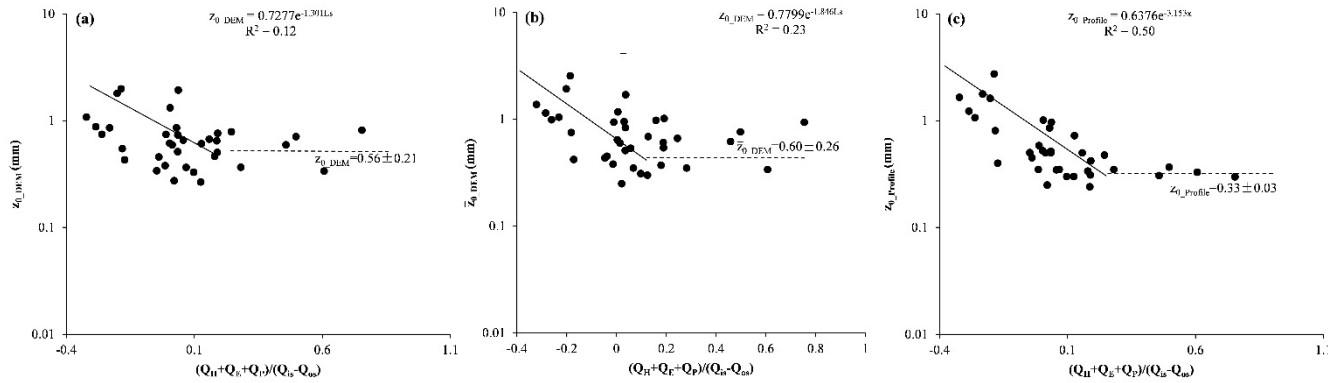

Figure 12. Aerodynamic surface roughness vs. $L_S$. Where $L_S=(Q_H+Q_E+Q_P)/(Q_{is}-Q_{os})$, in Figure 12(a) $z_{0\_DEM}$ was estimated based on prevailing upwind direction DEM based, in Figure 12(b) $\bar{z}_{0\_DEM}$ was the average of four cardinal wind directions $z_0$ to represent overall aerodynamic surface roughness, in Figure 12(c) $z_{0\_Profile}$ was the average of two orthogonal directions $z_0$.

Table 1 Measurement specifications for the AWS located at the top of the glacier (4820 m a.s.l.). The heights indicate the initial sensor distances to the glacier surface; the actual distances derived from the SR50A sensor.

| Variable | Sensors | Stated accuracy | Initial Height (m) |
|---|---|---|---|
| Air temperature | Vaisala HMP 155A | ± 0.2ºC | 2, 4 |
| Relative humidity | Vaisala HMP 155A | ± 2% | 2, 4 |
| Wind speed | Young 05103 | ± 0.3 m/s | 2, 4 |
| Wind direction | Young 05103 | ± 0.3º | 2, 4 |
| Ice temperature | Apogee SI-11 | ± 0.2ºC | 2 |
| Shortwave radiations | Kipp&Zonen CNR-4 total | ± 10% day | 2 |
| Longwave radiation | Kipp&Zonen CNR-4 total | ± 10% day | 2 |
| Surface elevation changes | Campbell SR50A | ± 0.01 m | 2 |
| Precipitation | OTT Pluvio$^2$ | ± 0.1 mm | 1.7 |

Table 2 Control point RMSE for manual and automatic photogrammetry

| Ground control points | | X error (mm) | Y error (mm) | Z error (mm) | Total error (mm) |
|---|---|---|---|---|---|
| Automatic | Point 1 | 0.71 | 5.83 | 6.61 | 5.11 |
| | Point 2 | 0.41 | 1.14 | 0.74 | 0.82 |
| | Point 3 | 0.54 | 4.55 | 2.40 | 2.99 |
| | Point 4 | 0.45 | 0.76 | 1.04 | 0.79 |
| | **Average** | **0.54** | **3.76** | **3.58** | **3.01** |
| Manual | Point 2 | 0.62 | 0.43 | 0.81 | 1.11 |
| | Point 4 | 0.44 | 0.27 | 0.43 | 0.67 |
| | Point 5 | 0.18 | 0.47 | 0.85 | 0.99 |
| | Point 7 | 0.66 | 0.39 | 2.97 | 3.07 |
| | **Average** | **0.52** | **0.40** | **1.65** | **1.78** |

**Table 3 Check point RMSE for manual and automatic photogrammetry**

| Ground Check points | | X error (mm) | Y error (mm) | Z error (mm) | Total error (mm) |
|---|---|---|---|---|---|
| Automatic | Point 5 | 2.06 | 4.44 | 7.70 | 5.27 |
| | Point 6 | 0.91 | 3.56 | 1.95 | 2.40 |
| | Point 7 | 0.98 | 3.11 | 2.60 | 2.41 |
| | **Average** | **1.41** | **3.74** | **4.83** | **3.62** |
| Manual | Point 1 | 0.30 | 0.19 | 0.39 | 0.52 |
| | Point 3 | 0.79 | 0.37 | 0.69 | 1.12 |
| | Point 6 | 0.28 | 0.83 | 0.90 | 1.26 |
| | Point8 | 0.46 | 0.45 | 0.44 | 0.77 |
| | **Average** | **0.52** | **0.53** | **0.66** | **0.99** |

**Table 4 The lagged correlation between $z_0$ and the main energy items during the melting season, the sensible heat and latent heat here was calculated based on the bulk method.**

| $z_{0\_Profile}$ | n | $(Q_{is}-Q_{os})$ | $Q_L$ | $Q_E$ | $Q_H$ | $L_S$ |
|---|---|---|---|---|---|---|
| Lag-0 | 64 | 0.143 | -0.309* | -0.614* | -0.088 | -0.578* |
| Lag-1 | 63 | 0.131 | -0.346* | -0.646* | -0.137 | -0.572* |
| Lag-2 | 62 | -0.022 | -0.113 | -0.356* | -0.307* | -0.585* |
| Lag-3 | 61 | -0.144 | 0.051 | -0.193* | -0.283* | -0.523* |
| Lag-4 | 60 | -0.142 | -0.241 | -0.016 | -0.013 | -0.205 |

n= the number of samples, *P<0.05