# Peer review of "Spatial and temporal variations in glacier aerodynamic surface roughness during the melting season, as estimated at August-one ice cap, Qilian mountains, China"

_The Cryosphere, 2019_

## Referee Comment (RC1) · Joshua Chambers (Referee) · 29 Oct 2019

General comments:

In this study, which is well within the remit of the journal, the authors present some interesting, hard-won (by the sounds of it) microtopographic and meteorological data from the August-one ice cap, China. They implement novel methods to collect some of their photogrammetric data automatically, in a location that is underrepresented in the glaciological literature.

[Figure]

Methods and data are presented and explained reasonably clearly, with some valuable insights given through comparison between microtopographic and meteorological measurements. While there is no independent validation of z0 values with other methods of obtaining z0 (wind profiles, eddy covariance), this is one of few studies that shows how the microtopographic methods used here can produce sensible values for melt volumes in the wider context of glacier monitoring. The temporal aspect of the work is a worthwhile inclusion, not just for the interesting nature of the data, but for the implications if such patterns were observed/studied elsewhere.

Overall it is well written and structured logically, and does not need much revision to make it publishable. Suggestions are fairly minor, although I would suggest that: 1) some terminology should be adjusted (see specific comments regarding 'surface roughness', 'direct measurement' etc ), 2) methods need further justification, in that some additional studies should be read/cited (again, see specific comments) and 3) figures could be of higher quality generally (i.e. do not just use screenshots for compound figures).

Specific comments:

Abstract Seeing as your work relates to z0 and not albedo, I would remove the mentions of albedo from the abstract to avoid confusion.

Introduction

Line 32: here, and throughout the manuscript, make sure to add a space between citations listed in parentheses and separated by semi-colons.

Line 41 – missed references to more recent studies using wind profiles:

Miles, E.S., Steiner, J.F. and Brun, F., (2017). Highly variable aerodynamic roughness length (z0) for a hummocky debris-covered glacier. Journal of Geophysical Research: Atmospheres, 122(16), pp.8447-8466.

Quincey, D., Smith, M., Rounce, D., Ross, A., King, O. and Watson, C., (2017). Evaluating morphological estimates of the aerodynamic roughness of debris covered glacier ice. Earth Surface Processes and Landforms, 42(15), pp.2541-2553.

Line 42 – "direct measurement of z0 has been shown to be more accurate than previous methods" – it is unclear what methods are referred to by this statement. Wind profile and microtopographic values are both estimates based on models. Please clarify or correct, and make sure it is clear throughout the rest of the paper that microtopographic z0 is an estimate, not a measurement.

Line 44 – "Current research has increasingly used direct measurement." Terminology needs adjusting to reflect the previous comment.

Line 47 – as above.

Line 49 – 51: The first sentence could be backed-up by several examples including Irvine-Fynn et al (2014), Smith et al (2016), Quincey et al (2017), Miles et al (2017), and Fitzpatrick et al (2019). The second and third sentences are confusing; while Kääb and Vollmer (2000) utilised aerial photography for photogrammetry, this was not used for a purpose related to ice roughness. The next sentence "Digital photos were taken against a dark background plate" does not refer to a part of the cited study, but rather to Rees (1999), who published the method mentioned.

Data and methods – overall this is very clear, and the photogrammetry details are nice to see.

Line 72: it would be interesting and useful background to include some information on the normal influence of the turbulent fluxes at this location.

Figure 1: Some scale would be useful in both panels. Is the figure a screenshot? Some artefacts have made their way into the top of the figure. Also some place names for context in panel (a) would help.

Line 93-94: Figure 2b does not illustrate the frame very well, in fact it is quite unclear what the image shows.

Line 99: in which direction did the camera move? Along the frame, or into it?

Line 117: what was the rationale for the plot size?

Figure 2: do you have any other site photos? Panel (b) is not very useful as it is, and some detail is not shown by panel (3).

Line 131: it might be useful to refer to the work of James & Robson (2014) and James et al (2017) for some critiques of using Agisoft Photoscan.

Line 149: repetition of reference.

Line 156: Smith et al (2016) calculated h* from the mean vertical extent above a de-trended plane. Hopefully this important step has just been omitted from the text (in which case it should be added, as detrending is a vital part of the method), and not from your calculations.

Line 162: please reference Munro (1989) for the profile-based simplification of the Lettau (1969) equation.

Line 174: Fitzpatrick et al (2019) also provide useful discussion of microtopographic methods. In addition, please clarify terminology – I would suggest reconsidering the use of the term 'surface roughness' as it can refer to one of a number of metrics (Smith, 2014), and could be more specific.

Results

Section 3.1 Photogrammetry precision: while this is important to report, much of the text is summarised in the two tables and two figures. If you were looking to cut down on text, perhaps this section could be more concise.

Line 213: change geo-reference to geo-referencing. Also, I'm not sure which value is being referred to by saying that "errors were less than 1 millimeter", as most of the averages in the tables are >1 mm.

Line 216: define RMSE before the first use of the acronym (line 213), not after the second time.

Line 227: Note that the accuracy requirements given by Rees and Arnold (2006) were for 2D topographic transects, not 3D plots.

Line 237: change 'covered' to 'covering'

Line 237: "z0 was highly variable" – it's worth keeping some perspective here. While z0 varied, it did so by less than 3 mm.

Figure 5: There is a typo on the y-axis label which should read 'surface roughness'. Also please see my previous note on using the term 'surface roughness'.

Line 258: Should be 'both of which occurred in periods of transition'.

Line 261: This is an interesting finding. Can you provide more detail? Can you include the actual values for the manually collected data that show the same pattern? Additionally, in the methods it is mentioned that z0 is an average of all four directional values – were the individual values analysed for directional influence?

Line 265: While z0 certainly changed over time, I do not think it is correct to say that it was related to the date. It was different when measured on different days, but this is because of factors other than what day of the month it is.

Line 268: is the 'terminal' the same as the terminus of the glacier? The latter expression is more commonly used.

Line 269: Change to 'At higher altitudes'

Line 275: Please be more specific than just saying "Manual investigation" – I take it here you are referring to photogrammetric data collected manually?

Lines 306-309: I am not sure that a separate introduction is required here. The final two sentences could be tacked onto the beginning of the next paragraph.

Line 335: changed "account" to "accounted".

Line 360: the r2 value reported here is different to the one shown in Figure 9. This is also the case for line 370 and fig. 11a, and line 372/fig. 11b.

Discussion

Line 412: I do not think there needs to be a summary here – all of the information should be apparent from the main text.

Line 414: Do not need to cite these again here.

Line 416: I notice that the difference between ice z0 and snow z0 is very small. Can you comment on this in the text? Some find that the difference can be an order of magnitude. Were both surfaces at your site particularly smooth? Or could it be something to do with the size of the patch (thinking about the scale/resolution dependency of the microtopographic method – see Fitzpatrick et al. 2019).

Lines 422-425: this paragraph needs rewording so that the first sentence does not seem disconnected from the rest.

Lines 430-433: this is a significant finding; however, there is something about the wording in this sentence that I think should be addressed – as z0 is in this instance (using the bulk method) required to calculate the turbulent fluxes, arguing that the turbulent heat index (calculated with turbulent fluxes) is a determining factor seems circular. I think the statement could be made more clearly, perhaps referring to the association between the two rather than a causal relationship.

Line 434: Make sure terminology is clear here – you refer to the August-one ice cap, and then call it a glacier. In my understanding, these are different.

Line 439: The second sentence can be deleted, it does not add anything to the findings or argument.

Conclusion

I think comparison to other ice masses, and links to other studies/locations should be made in the discussion, with some thought given to whether you might find the same results where ice z0 and snow z0 have greater contrast. And, while it is important to acknowledge the site specificity of a study, further studies are always required and saying so in the conclusions is superfluous. Instead, the main messages from the paper (3 or 4 of them, as far as I can see) should be summarised here.

References cited in comments:

Fitzpatrick, N., Radić, V., & Menounos, B. (2019). A multi-season investigation of glacier surface roughness lengths through in situ and remote observation. The Cryosphere, 13(3), 1051-1071.

Irvine-Fynn, T., Sanz-Ablanedo, E., Rutter, N., Smith, M., & Chandler, J. (2014). Measuring glacier surface roughness using plot-scale, close-range digital photogrammetry. Journal of Glaciology, 60(223), 957-969.

James, M. R., & Robson, S. (2014). Mitigating systematic error in topographic models derived from UAV and ground‐based image networks. Earth Surface Processes and Landforms, 39(10), 1413-1420.

James, M. R., Robson, S., & Smith, M. W. (2017). 3‐D uncertainty‐based topographic change detection with structure‐from‐motion photogrammetry: precision maps for ground control and directly georeferenced surveys. Earth Surface Processes and Landforms, 42(12), 1769-1788.

Kääb, A., & Vollmer, M. (2000). Surface geometry, thickness changes and flow fields on creeping mountain permafrost: automatic extraction by digital image analysis. Permafrost and Periglacial Processes, 11(4), 315-326.

Lettau, H. (1969). Note on aerodynamic roughness-parameter estimation on the basis of roughness-element description. Journal of applied meteorology, 8(5), 828-832.

Munro, D. S. (1989). Surface roughness and bulk heat transfer on a glacier: compari-

son with eddy correlation. Journal of Glaciology, 35(121), 343-348.

Rees, W. G. (1998). A rapid method of measuring snow-surface profiles. Journal of Glaciology, 44(148), 674-675.

Rees, W. G., & Arnold, N. S. (2006). Scale-dependent roughness of a glacier surface: implications for radar backscatter and aerodynamic roughness modelling. Journal of Glaciology, 52(177), 214-222.

Smith, M. W. (2014). Roughness in the earth sciences. Earth-Science Reviews, 136, 202-225.

Smith, M. W., Quincey, D. J., Dixon, T., Bingham, R. G., Carrivick, J. L., Irvine‐Fynn, T. D., & Rippin, D. M. (2016). Aerodynamic roughness of glacial ice surfaces derived from high‐resolution topographic data. Journal of Geophysical Research: Earth Surface, 121(4), 748-766.

---

## Referee Comment (RC2) · Evan Miles (Referee) · 29 Oct 2019

**Review of 'Spatial and temporal variations in glacier surface roughness during melting season, as observed at August-one, Qilian mountains, China' by J Liu, R Chen, and C Han**

The study by J Liu et al. demonstrates the first use of an automated photogrammetric apparatus to monitor surface roughness at a daily timescale for an ice cap in China. The authors supplement these observations from a single site with meteorological records from nearby, as well as manual photogrammetric measurements at a variety of locations across the ice cap during the course of the ablation season. The authors thus investigate spatial and temporal variations of surface roughness during the ablation season, as well as linkages to surface energy balance. From the automated roughness measurements they find that roughness is temporally variable and highly modified by precipitation, with both rain and snow precipitation leading to a reduction in roughness. From the manual measurements, they find that the seasonal firm/ice transition zone corresponds to the maximal surface roughness at any point, while ice or snow surfaces both exhibit lower surface roughness. The authors also suggest a link to the importance of turbulent fluxes in the whole energy balance.

The target of spatially-extensive surface roughness measurements is a novel development, and useful to understand roughness variations. While the general patterns of seasonal and spatial variability are very likely to be accurate and form a nice story, the authors seem to have some fundamental misunderstandings about surface roughness metrics and their meaning. In addition, the methods are not entirely clear, results are given to an unrealistic and misleading precision (also without any uncertainty assessment), and although the written English is generally correct, the writing style is particularly abrupt. Consequently, although the authors have painted a nice picture of the spatiotemporal evolution of surface roughness at August-one ice cap, the manuscript needs substantial revisions before it should be considered for publication in The Cryospehere.

*Major points:*

Fundamental misunderstanding of surface roughness. The authors seems to confuse $Z_o$ and topographic surface roughness, which are not the same: while approaches have linked the two, the aerodynamic roughness length is not simply a topographic parameter, and efforts to assess $Z_o$ based on topographic parameters need to be validated with micrometeorological measurements. Furthermore, the authors' effort to produce a grid-based estimate of surface roughness is only applicable for the case of isotropic roughness, which is not the case for ice surfaces.

Lack of clarity with regards to several methods. The authors mention two specific efforts to estimate $Z_o$ from topographic profiles: Lettau (1969) and Munro (1989). It is not clear which is actually used in this study, now how it was applied to the gridded height data. In addition, numerous details of the energy balance model used are missing, while the authors may have accidentally disregarded conduction of heat.

Unrealistic precision, no uncertainty of $Z_o$ estimates or energy balance. The accuracy of $Z_o$ is provided relative to control and check points on photogrammetric frames, and is reported to the tenth of a millimetre. However, it is unlikely that the actual measured positions of their control and check points are known to this accuracy. Furthermore, the surface height models produced by the structure-from-motion processing appear to be oversampled by a factor of 10x in each dimension,

relative to the reported point densities. Finally, no assessment of uncertainty has been conducted for the $Z_o$ estimates or the energy balance calculations.

No evidence given of cryoconite, but of red algae. This may be a misunderstanding of some sort, but the authors refer multiple times to the development of cryoconite and its effect on surface roughness, a phenomenon that would certainly explain some of the surface roughness dynamics that they observe. However, the first time cryoconite is mentioned is with regards to Figure 2, but Figure 2 does not provide any evidence (to my eye) of cryoconite – rather, red snow algae is clearly evident. This gives some concern of a basic misinterpretation of results.

Some grammar improvement needed, also some changes to the writing style are needed, as it is not currently suitable for TC.

*Detailed comments:*

L1-2…during 'the' melting season

L18. Zo was calculated from this data – you need to say how. Manual measurements of what type? Micrometeorology? Profiles of elevation difference?

L37-63. It is apparent from this section that the authors misunderstand several key concepts relating to Zo and turbulent heat transfer more generally; I suggest a careful read of Smith et al (2016) for a review of the differences. First, $Z_o$ may be commonly called 'surface roughness' but its full title is the 'aerodynamic roughness length' (for momentum transfer/heat transfer). In any case, it emerges in the bulk aerodynamic approach as a constant of integration that results from the interaction of the boundary layer with the surface. It is a meteorological term (not a topographical term) that is influenced by both properties of the boundary layer and the surface. One can determine an effective surface roughness 'directly' from eddy covariance measurements (and less directly from wind towers), but it is highly variable in time primarily because the boundary layer is often highly variable. The variability of the boundary layer leads to a different fetch over which the layer is interacting with the surface topography. The microtopographic roughness (which you have calculated) is thus a very good indication of $Z_o$, but the relationship is not direct or linear, as the energy balance is controlled not just by surface topography at an individual location, but is variably influenced by its surroundings (e.g. Steiner et al, 2019). Thus, it is difficult to trust the values of $Z_o$ produced by this study, as they are not validated by wind tower or eddy covariance observations (which actually resolve $Z_o$). However, microtopographic roughness metrics are a very strong proxy for $Z_o$ (e.g. Nield et al, 2013), so I have much more confidence in the temporal and spatial variability presented by the authors. However, I think they need to very carefully reframe their introduction to conform with established theory.

L42. Please provide references indicating that microtopographic $Z_o$ is more accurate than wind profile or EC measurements. I don't know how one can claim this, as those methods are the 'ground truth' of $Z_o$ at a site.

L47. 'Direct measurement' is strange nomenclature; microtopographic approaches, including the Lettau (1969) approach, are anything but direct.

L52. Rees and Arnold (2006) is also sensible to mention here.

L55. Other examples of this approach are Rounce et al (2015), Quincey et al (2017), and Miles et al (2017).

L56. The photogrammetric approaches need validation, as the relationship between topographic roughness and aerodynamic roughness length is also affected by local meteorology (Nield et al, 2013).

L74/Figure1. Both panels need a scale. The political map of China is irrelevant to the current study; of more relevance are dominant weather patterns and elevation, including areas outside China's claimed border. Furthermore there is no need to depict the South China Sea, which results in a very poor use of space. What is the polygon within China? It is not identified in the figure or caption. Please provide information about the image of August-one in panel (b) – date, satellite, etc. Red and green are poor choices of color for icons in panel (b), as many people cannot distinguish between these two colors.

L79. Please provide sensor specifications and measurement uncertainties for the AWS.

L81. The sensor measures relative surface height; it does not measure mass balance. Also in L104

L83. There was a windbreak fence installed on the glacier?

L94. How were the positions of the control and check points measured? You report accuracies relative to these positions of less than 1 mm, but I am not convinced that you could locate the control point positions to a higher accuracy than this. Also, how was the frame structure anchored?

L102. Did you choose the daily best-exposed sets of photos manually or automatically? For days with multiple very clear photo sets, was there strong agreement in derived $Z_o$ or a consistent diurnal variation?

L112. Does the August-one ice cap have an accumulation area?

L120. How are the seven pairs of convergent photos arranged? Do you use all 14 photos to produce the DEM and orthoimage? Did you ever carry out the manual photogrammetry at the automatic site?

L124/Figure 4. Panels b and c are switched relative to the text, which led to some confusion about the numbers of check points and control points. I see no evidence of cryoconite in the image, but of red algae which is commonly found on melting snow.

L135. The standard reference for this processing workflow as applied to glaciers is Westoby et al (2012). Also, this approach has already been applied to estimate surface roughness of glacier surfaces: Quincey et al (2017), Miles et al (2017), Steiner et al (2019).

L141-146. This content belongs in the background. Note that Lettau (1969) was the first such effort (of which I am aware). It is also worth noting the extensive review of microtopographic metrics by Nield et al (2013).

L145. Munro (1989) is probably the appropriate first reference here, as is Brock et al (2006).

L161. The method described (based on the standard deviation of detrended elevation) is precisely the Munro (1989) method.

L172-3. Averaging over cardinal directions is only meaningful for surfaces that are isotropic. However, the literature has repeatedly shown that melting ice is strongly anisotropic, as the direction of wind strongly dictates the pattern of melt, and feeds back via roughness. So this 'averaging all cardinal profiles' is entirely unsuited to your study site, unless you can demonstrate that the ice surface is indeed isotropic in terms of roughness, which would be highly surprising.

L174. Some things are not entirely clear to me about your method. First, do you use all profiles in each cardinal direction? Second, it is not clear if you have implemented the exact Lettau approach or the Munro approximation in your 'all profiles' approach. Third, such an implementation (all profiles averaged, for either Lettau or Munro) has already been implemented and tested for a glacier surface. Please see Miles et al, (2017).

L179-181. The surface energy balance presented is not quite accurate for a 'melting' glacier, but for a 'temperate' glacier. Do you have any evidence that the August-one ice cap is temperate? If not, there also needs to be a term for heat conduction.

L191. My impression is that you use your calculated $Z_o$ value for the bulk aerodynamic approach. How do you integrate your 3-hourly (half-day) $Z_o$ values with your model? At what timescale is the model run? What uncertainty does the input meteorology have, and what uncertainty does this produce for your results?

L204. Is an environmental lapse rate entirely appropriate for this site? Do you have lapse rate measurements?

L205. How confident are you that the AWS measurements are broadly representative of the entire ice cap? Do you have evidence to back up this claim?

L210-211. I am not sure how you get seventeen (17), as you have 4 control points and 3 check points. Similarly, I do not understand what the 31 manual photography pairs are – please explain.

L210-219. This entire section is an amalgamation of bullet points; please rewrite to conform to style for The Cryosphere.

L212 and L216. The reported point densities do not justify a resolution of 0.1mm, but of 1mm. These DEMs are 100x oversampled.

L213. The average georeferenced error is greater than 1mm for half of the control points, and nearly all check points. However, I am also not certain how precisely you could have measured the location of the control and check points. Please provide details and uncertainty.

L225. Yes, but part of this is also the difference of your survey design. For the automatic measurements, the camera is moving linearly, and the density of tie-points is much higher in the foreground compared to the background. For the manual method, although the survey design is not clear, more photos were taken and I presume that they surrounded the target area. This type of survey would be expected to provide a much more robust elevation model.

L228. Rees and Arnold (2006) did indeed suggest millimetre vertical accuracy. The also suggest a fetch length of 3-6 meters as relevant for the majority of energy balance situations, which is considerably larger than your domain.

L230/Figure3. It is not clear what this chart shows – the y axis is labelled 'Differences', but is this RMSE, MAD, or…? Please clarify.

L231-4/Figure 4. Same problem and Figure 3. Should be merged with Figure 3 as a second panel.

L238. No description of profile analysis is included in the manuscript, only of a DEM analysis. Please provide more detail.

L239. Do you have an estimate of the uncertainty of these $Z_o$ values?

L242-254. Listing a narrative as bullet points in the results is not particularly aesthetic, and this section should be rewritten as a paragraph. More importantly, this section mixes results and interpretations. Please present the observations, then interpret them.

L254/Figure 5. $Z_o$ values are more commonly presented on a logarithmic scale, as even a factor of 2 makes little difference in the turbulent fluxes, whereas a factor of 10 can be a considerable difference regardless of value. This is, in part, due to the bulk aerodynamic approach. Also, it would be very nice to include a set of panels depicting the surface at different parts of this record (high and low values, for example).

L258. One order of magnitude is not a particularly large variation of $Z_o$.

L260. I have not yet seen evidence of cryoconite holes; the image in Figure 2 is unconvincing. Also applies to L280

L263,276,277. I see no need to include p-values here.

L274. Was there no accumulation in this year?

L283/Figure 6. Is there a reason that the lines are shown with different styles? For comparison, it would be good for all 4 panels to have the same y-axis limits.

L288-302. Somewhere in this section there should be a reference to Figure 7.

L310,324,353,339,345. The use of sub-headings here just breaks up the text.

L320/Figure 7. In panel (a), please use a logarithmic scale for $Z_o$. Is panel (c) showing net solar radiation, or downwelling – not specified. The y-axis upper limit in panel (d) should be 100%. In general, all time-series look smoothed. Please provide details of exactly what is shown. In the caption, please be sure to provide the year.

L330/Figure 8. What is the uncertainty of each of these values quantities?

L335. If latent heat and sensible heat account for so little of the energy balance, how much impact does a variation of $Z_o$ from 0.25 mm to 2.5 mm have on the total energy balance?

L343. The 'visible smoothing' is not clear to me from Figure 7. Please explain where you see this.

L349-350. As turbulent fluxes matter very little for your energy balance, the match is not due to the calculated $Z_o$.

L358-380/Fig10 and 11. I do not think this analysis is very well grounded in theory. First of all, as the turbulent fluxes depend on $Z_o$, you are comparing a quantity to a modified version of itself in Figure 10d and Figure 11. In fact, this exactly corresponds to the shape of the fit in bulk aerodynamic theory (which you have used to relate $Z_o$ to the turbulent fluxes). So on one hand, none of this section is unexpected, but nor does it provide any novel insight. On the other hand, if you intend to examine the potential feedbacks between energy balance and surface roughness, that would be very interesting, but would require the use of a lagged correlation (in which case your variables would be independent).

L389. Again, Westoby et al (2012) is probably an even more appropriate reference here.

L391. I disagree with this because your survey setup is entirely different for the manual and automatic methods. See my comment with regards to L225.

L400. I believe you are referring to the glacier terminus. Please replace 'terminal' with 'terminus' throughout the manuscript.

L403. This is the very interesting result of your study: following the zone of maximum roughness as it migrates upglacier. But a key question is how important are turbulent fluxes in this zone? Perhaps they are relatively unimportant everywhere else, but in this transition zone you have maximum $Z_o$ and the zone also migrates across much of the glacier, highlighting the importance of transient surface characteristics.

L429. Please be careful and consistent with the terminology that you use. In this study you have examined topographic roughness and the aerodynamic roughness length (which are not quite the same thing, see Smith et al, 2016).

L431. I do not think this is a meaningful result, see my comment on L358-380. This also applies to L439.

L434. What do you mean by 'heavy-loading glacier'? I have not heard the term before.

L437. The link between cryoconite holes and surface roughness is indeed important, and you should make this link explicit earlier. However, your manuscript has not presented any clear evidence of the cryoconite development process occurring at your site.

L440. I do not understand what you are referring to here, with regards to quantitative vs qualitative research. Please explain more clearly what you are implying.

L456. What type of studies? Please make some concrete suggestions; at present this discussion and conclusion makes very little contribution to the field.

L470 and L472. Duplicate reference.

*References mentioned in the review*

Miles et al. (2017). Highly variable aerodynamic roughness length (z0) for a hummocky debris-covered glacier. *Journal of Geophysical Research: Atmospheres*

Nield et al. (2013). Estimating aerodynamic roughness over complex surface terrain. *Journal of Geophysical Research: Atmospheres*

Quincey et al., (2016). Evaluating morphological estimates of the aerodynamic roughness of debris-covered glacier ice. *Earth Surface Processes and Landforms*

Rounce et al., (2015). Debris-covered glacier energy balance model for Imja-Lhotse Shar Glacier in the Everest region of Nepal. *The Cryosphere*

Steiner et al., (2018). The Importance of Turbulent Fluxes in the Surface Energy Balance of a Debris-Covered Glacier in the Himalayas. *Frontiers in Earth Science*

Westoby et al., (2012). 'Structure-from-motion' photogrammetry : A low-cost, effective tool for geoscience applications. *Geomorphology*

---

## Author Comment (AC3) · 3 Jan 2020

The comment was uploaded in the form of a supplement:
https://www.the-cryosphere-discuss.net/tc-2019-186/tc-2019-186-AC3-supplement.pdf

**Fig. 1.**

[Figure]

Fig. 2.

[Figure]

**Fig. 3.**

---

## Author Comment (AC4) · 3 Jan 2020

The comment was uploaded in the form of a supplement:
https://www.the-cryosphere-discuss.net/tc-2019-186/tc-2019-186-AC4-supplement.pdf

[Figure]

**Fig. 1.**

(a)

(b)

DEM
2.8(cm)

-3.0(cm)
0 10 20   40
cm

(c)

● Check point
▲ Control point

(d)

DEM
27.5 (cm)

-15.8 (cm)
0 5 10  20  30  40
cm

**Fig. 2.**

[Figure]

**Fig. 3.**

[Figure]

**Fig. 4.**

[Figure]

[Figure]

**Fig. 5.**

**Fig. 6.**

[Figure]

**Fig. 7.**

[Figure]

**Fig. 8.**

[Figure]

**Fig. 9.**

[Figure]

**Fig. 10.**

**(a)** $y = 0.0005x + 0.6477$
$R^2 = 0.0058$

$z_{0\_DEM}$ (mm)

$(Q_{is}-Q_{os})$ (Wm$^{-2}$)

**(b)** $y = -0.0048x + 0.5259$
$R^2 = 0.1218$

$Q_L$ (Wm$^{-2}$)

**(c)** $y = 0.0017x + 0.6894$
$R^2 = 0.0036$

$z_{0\_DEM}$ (mm)

$Q_H$ (Wm$^{-2}$)

**(d)** $y = 0.5575e^{-0.008x}$
$R^2 = 0.1207$

$Q_E$ (Wm$^{-2}$)

**Fig. 11.**

[Figure]

**Fig. 12.**

---

## Author Response (AR1)

**Spatial and temporal variations in glacier aerodynamic surface roughness during the melting season, as estimated at August-one ice cap, Qilian mountains, China**

Junfeng Liu et al.

Dear Editor,

We have carefully revised the manuscript according to the comments from referee #1 and #2. The most important comments are that 1) misunderstanding of surface roughness with aerodynamic surface roughness; 2) unclear of method part; 3) precision problem of manual and automatic photogrammetry; 4) ice surface is cryoconite or red snow algae? 5) Figures does not meet the high quality standards of TC.

We have given some carful explanations in our reply;please see the detailed point-by-point responses below. The corresponding changes have been made in the revised paper, track changes was used in order to be easily identified. Marked-up manuscript was given at the end of the replies. We hope the revised manuscript is suitable for the journal.

Best regards,

Junfeng Liu

**Reply to comments from referee Joshua Chambers**

General comments:

In this study, which is well within the remit of the journal, the authors present some interesting, hard-won (by the sounds of it) microtopographic and meteorological data from the August-one ice cap, China. They implement novel methods to collect some of their photogrammetric data automatically, in a location that is underrepresented in the glaciological literature.

Methods and data are presented and explained reasonably clearly, with some valuable insights given through comparison between microtopographic and meteorological measurements. While there is no independent validation of z0 values with other methods of obtaining z0 (wind profiles, eddy covariance), this is one of few studies that shows how the microtopographic methods used here can produce sensible values for melt volumes in the wider context of glacier monitoring. The temporal aspect of the work is a worthwhile inclusion, not just for the interesting nature of the data, but for the implications if such patterns were observed/studied elsewhere.

Overall it is well written and structured logically, and does not need much revision to make it publishable. Suggestions are fairly minor, although I would suggest that:

1) some terminology should be adjusted (see specific comments regarding 'surface roughness', 'direct measurement' etc ), 2) methods need further justification, in that some additional studies should be read/cited (again, see specific comments) and 3) figures could be of higher quality generally (i.e. do not just use screenshots for com- pound figures).

**Reply: In the revised version of the paper, we adjusted the terminology of 'surface roughness' as 'aerodynamic surface roughness'. In the methods part, we cite these latest studies. Figures in the revised version have higher quality.**

Specific comments:

Abstract seeing as your work relates to z0 and not albedo, I would remove the mentions of albedo from the abstract to avoid confusion.

**Reply: We delete accordingly.**

Introduction

Line 32: here, and throughout the manuscript, make sure to add a space between citations listed in parentheses and separated by semi-colons.

**Reply: Done**

Line 41 – missed references to more recent studies using wind profiles:

Miles, E.S., Steiner, J.F. and Brun, F., (2017). Highly variable aerodynamic roughness length (z0) for a hummocky debris-covered glacier. Journal of Geophysical Research: Atmospheres, 122(16), pp.8447-8466.

Quincey, D., Smith, M., Rounce, D., Ross, A., King, O. and Watson, C., (2017). Evaluating morphological estimates of the aerodynamic roughness of debris covered glacier ice. Earth Surface Processes and Landforms, 42(15), pp.2541-2553.

**Reply: Thanks for your suggestions, we cited related literatures as**

**Miles, E.S., Steiner, J.F. and Brun, F., (2017). Highly variable aerodynamic roughness length (z0) for a hummocky debris-covered glacier. Journal of Geophysical Research: Atmospheres, 122(16), pp.8447-8466. doi:10.1002/2017JD026510**

**Quincey, D., Smith, M., Rounce, D., Ross, A., King, O. and Watson, C., (2017). Evaluating morphological estimates of the aerodynamic roughness of debris covered glacier ice. Earth Surface Processes and Landforms, 42(15), 2541-2553. DOI:10.1002/esp.4198.**

Line 42 – "direct measurement of z0 has been shown to be more accurate than previ- ous methods" – it is unclear what methods are referred to by this statement. Wind pro- file and microtopographic values are both estimates based on models. Please clarify or correct, and make sure it is clear throughout the rest of the paper that microtopographic z0 is an estimate, not a measurement.

**Reply: Thanks for your suggestions. We delete the sentence in Line 42 and rewrite as "Glacier surface $z_0$ has been widely studied through methods such as eddy covariance (Munro, 1989; Smeets et al., 2000; Smeets and Van den Broeke, 2008; Fitzpatrick et al., 2019), or wind profile (Wendler and Streten, 1969; Greuell and Smeets, 2001; Denby and Snellen, 2002; Miles et al., 2017; Quincey et al., 2017). However, micro-topographic estimated $z_0$ shows some advantages, such as lower scatter, rather than profile measurements over slush and ice (Brock et al., 2006), and ease of application at different locations (Smith et al., 2016)."**

**The "direct measurement" changed to "microtopographic estimated z0". The rest of the paper also changed accordingly.**

Line 44 – "Current research has increasingly used direct measurement." Terminology needs adjusting to reflect the previous comment.

**Reply: Done.**

Line 47 – as above.

**Reply: Done.**

Line 49 – 51: The first sentence could be backed-up by several examples including Irvine-Fynn et al (2014), Smith et al (2016), Quincey et al (2017), Miles et al (2017), and Fitzpatrick et al (2019). The second and third sentences are confusing; while Kääb and Vollmer (2000) utilised aerial photography for photogrammetry, this was not used for a purpose related to ice roughness. The next sentence "Digital photos were taken against a dark background plate" does not refer to a part of the cited study, but rather to Rees (1999), who published the method mentioned.

**Reply: We added these references in the first sentence.**

**The following part has changed as 'Initially, the Micro-topographic method was developed as snow digital photos were taken against a dark background plate. The contrast between the surface photo and the plate could then be quantified as an estimation of surface roughness (Rees, 1998). This method is still widely applied to quantify glacier surface roughness (Rees and Arnold, 2006; Fassnacht et al., 2009a; Fassnacht et al., 2009b; Manninen et al., 2012).'**

Data and methods – overall this is very clear, and the photogrammetry details are nice to

see.

Line 72: it would be interesting and useful background to include some information on the normal influence of the turbulent fluxes at this location.

**Reply: we cited one published energy balance analysis results by Qing et al., (2018). The add part "Energy balance analysis indicated that net radiation contribute 86% and turbulent heat fluxes contribute about 14% to the energy budget in the melting season. A sustained period of positive turbulent latent flux exists on the August-one ice cap in August, causing faster melt rate in this period (Qing et al., 2018)."**

Figure 1: Some scale would be useful in both panels. Is the figure a screenshot? Some artefacts have made their way into the top of the figure. Also some place names for context in panel (a) would help.

**Reply: Done**

Line 93-94: Figure 2b does not illustrate the frame very well, in fact it is quite unclear what the image shows.

**Reply: we have revised accordingly.**

**In the revised manuscript, We split Figure 2 to Figure 2 and Figure3. Figure 2 showed the automatic photogrammetry. Figure 3 illustrate the automatic and manual photogrammetry control points and check points, the control frame, and the detrended DEM.**

Line 99: in which direction did the camera move? Along the frame, or into it?

**Reply :The camera was 1.7m above ice surface and move along the control frame.**

Line 117: what was the rationale for the plot size?

**Reply:Plots should large enough to include obstacles to represent the glacier**

**surface. The August-one glacier ice cap is generally smooth and uniform surface.**

**We expect the 1.1*1.1m plot is large enough to represent the dominant roughness**

**elements influencing z0. Additionally, the 1.1m*1.1m aluminum square is quite**

**portable and easily apply at different locations of glacier.**

Figure 2: do you have any other site photos? Panel (b) is not very useful as it is, and some detail is not shown by panel (3).

**Reply: we have used photo and corresponding DEM data to represent the manual and automatic photogrammetry acquired micro-scale surface roughness.**

Line 131: it might be useful to refer to the work of James & Robson (2014) and James et al (2017) for some critiques of using Agisoft Photoscan.

**Reply: Done**

**In this part, we cite James & Robson (2014) and James et al (2017) for some critiques of using Agisoft Photoscan. We also include two debris-covered glacier z0 estimation paper based on Agisoft.**

**The new paragraph rewrite as "Structure-from-motion photogrammetry is revolutionizing the collection of detailed topographic data (Westoby et al., 2012; James et al., 2017). High resolution DEMs produced from photographs acquired with consumer cameras need careful handling (James and Robson, 2014). In this study, both manual and automatically derived photographs were imported into a software program, Agisoft Photoscan Professional 1.4.0. This software allowed us to estimate camera intrinsic parameters, camera positions, and scene geometry. Agisoft Photoscan Professional is a commercial package which implements all stages of photogrammetric processing (James et al., 2017). It has previously been used to generate three-**

dimensional point clouds and digital elevation models of debris-covered glaciers (Miles et al., 2017; Quincey et al., 2017; Steiner et al., 2019), ice surfaces and braided meltwater rivers (Javernick et al., 2014; Smith et al., 2016). In our study, we found that after new snowfall, it was difficult to match feature points in the photo sets. Three days of automatic data could not be processed. We estimated $z_0$ data for the missing days based on data from snowfall days at the automatic site."

Line 149: repetition of reference.

**Reply:  Done**

Line 156: Smith et al (2016) calculated h* from the mean vertical extent above a de-trended plane. Hopefully this important step has just been omitted from the text (in which case it should be added, as detrending is a vital part of the method), and not from your calculations.

**Reply: For manual observation, the aluminum frame laid horizontally over the glacier surface. For automatic observation, the control field was also laid horizontally over the ice surface that lowered as the ice melted, and maintained a horizontally position between control field and ice surface. We have add the detrend method in line 565 as 'For manual photogrammetry, we put the aluminum frame horizontally over the ice surface, the plot is detrended by setting the control points at z axis of the same values. For automatic photogrammetry, the control field of wooden frame was also laid horizontally over the ice surface that lowered as the ice melted and maintained a horizontal position between the control field and ice surface. A DEM based approach enables the roughness frontal area $s$ to be calculated directly for each cardinal wind direction (Smith et al., 2016). The combined roughness frontal area was calculated across the plot, the ground area occupied by micro-topographic obstacles is $1m^2$. We used a DEM-based average ($\bar{z}_{0\_DEM}$) of four cardinal wind directions to represent overall aerodynamic surface roughness. Based on the half-hour wind direction data at the August-one ice cap, the daily upward wind direction DEM-based $z_{0\_DEM}$ was also estimated at the automatic photogrammetry site. Considering that wind direction changed during the day, in this case we selected the prevailing wind direction to calculate frontal area $s$. The prevailing upwind direction DEM-based $z_{0\_DEM}$ was applied to calculate turbulent heat flux. Using the Munro (1989) method, $z_{0\_Profile}$ was calculated for every profile (n=1000) in both orthogonal directions for each plot at the automatic photogrammetry site. '**

Line 162: please reference Munro (1989) for the profile-based simplification of the Lettau (1969) equation.

**Reply: Done,**

**In the revised manuscript, we not only apply Munro(1989) method but also calculate the z0 based upward wind direction DEM based z0 to represent the aerodynamic surface roughness, and applied to calculate turbulent heat flux.**

**We have revised in the manuscript as 'Based on the work of Lettau (1969), Munro (1989) simplified the equation (1) by assuming that h* can equal twice the standard deviation of elevations in the de-trended profile, with the profile's mean elevation set to 0 meter. The aerodynamic roughness length for a given profile then becomes '**

"

Line 174: Fitzpatrick et al (2019) also provide useful discussion of microtopographic methods. In addition, please clarify terminology – I would suggest reconsidering the use of the term 'surface roughness' as it can refer to one of a number of metrics (Smith, 2014), and could be more specific.

**Reply: Thanks for your recommendation about Fitzpatrick et al (2019) study about microtopographic methods, which have provide EC comparsion with DEM based z0 in multi-season . This paper also give detailed introduction   about z0 estimation from DEM.   We have referenced this paper in our study accordingly.**

**We add a sentence as line 152 as "Fitzpatrick et al. (2019) also developed two methods for the remote estimation of $z_0$ by utilizing lidar-derived DEM."**

**Consider the 'surface roughness' is not specific. We have revised the surface roughness as aerodynamic surface roughness in this paper.**

Results

Section 3.1 Photogrammetry precision: while this is important to report, much of the text is summarised in the two tables and two figures. If you were looking to cut down on text, perhaps this section could be more concise.

**Reply: we revised and provide uncertainty in the revised manuscript**

Line 213: change geo-reference to geo-referencing. Also, I'm not sure which value is being referred to by saying that "errors were less than 1 millimeter", as most of the averages in the tables are >1 mm.

**Reply: We agree, now the sentence is revised as "The average geo-referencing errors were fluctuate around 1 millimeter"**

Line 216: define RMSE before the first use of the acronym (line 213), not after the second time.

**Reply: We have changed accordingly.**

Line 227: Note that the accuracy requirements given by Rees and Arnold (2006) were for 2D topographic transects, not 3D plots.

**Reply: Thanks for remind, we delete this sentence accordingly.**

Line 237: change 'covered' to 'covering'

**Reply: Done**

**We have revised as' Data for ice surface roughness was collected from the automatic photogrammetry camera site from July 12 to September 15, a period covering the whole melting season.'**

Line 237: "z0 was highly variable" – it's worth keeping some perspective here. While z0 varied, it did so by less than 3 mm.

**Reply: We have revised accordingly.**

Figure 5: There is a typo on the y-axis label which should read 'surface roughness'. Also please see my previous note on using the term 'surface roughness'.

**Reply: We have changed as "Aerodynamic surface roughness"**

Line 258: Should be 'both of which occurred in periods of transition'.

**Reply: We have changed accordingly.**

Line 261: This is an interesting finding. Can you provide more detail? Can you include the actual values for the manually collected data that show the same pattern? Addition- ally, in the methods it is mentioned that z0 is an average of all four directional values – were the individual values analysed for directional influence?

**Reply: We did want analysis the four cardinal direction $z_0$ for manual data. But we did not strictly control the aluminum frame at certain direction during our field work at that time. We find at automatic site, at south to north direction $z_0$ seem larger than north to south direction $z_0$. We expect it is related with direct short wave radiation. We are not so sure. We need accumulate more field work to prove this.**

**In the revised manuscript of Figure 5, we have include DEM based four directions Z0 and prevailing wind direction z0. Munro profile method calculated z0 at two directions are also included.**

Line 265: While z0 certainly changed over time, I do not think it is correct to say that it was related to the date. It was different when measured on different days, but this is because of factors other than what day of the month it is.

**Reply: We totally agree. We have revised as 'Analysis indicated that $\bar{z}_{0\_DEM}$ proved to have an interesting relationship with altitude'**

Line 268: is the 'terminal' the same as the terminus of the glacier? The latter expres- sion is more commonly used.

**Reply: We have revised 'terminal' as 'terminus'**

Line 269: Change to 'At higher altitudes'

**Reply: Done**

Line 275: Please be more specific than just saying "Manual investigation" – I take it here you are referring to photogrammetric data collected manually?

**Reply: We totally agree, the sentence have revised as" Photogrammetric data collected manually revealed that ice surface roughness increased with altitude (Figure. 6c). From terminal to top, $z_0$ varied from 0.06 mm to 2.2 mm."**

Lines 306-309: I am not sure that a separate introduction is required here. The final two sentences could be tacked onto the beginning of the next paragraph.

**Reply: Agree. We have delete the first sentence and the final two sentences tacked onto the beginning of the next paragraph.**

Line 335: changed "account" to "accounted".

**Reply: Done**

Line 360: the r2 value reported here is different to the one shown in Figure 9. This is also the case for line 370 and fig. 11a, and line 372/fig. 11b.

**Reply: we showed r2 in In Figure 9, in line 360, we reported the correlation coefficient (r). In Figure 11a and Figure 11b, we also reported r2, and in line 370 we reported r instead of r2.**

**In the revised version, we reported r2 instead of r.**

Discussion

Line 412: I do not think there needs to be a summary here – all of the information should be apparent from the main text.

**Reply: We have revised the discussion part accordingly.**

Line 414: Do not need to cite these again here.

**Reply: Done**

Line 416: I notice that the difference between ice z0 and snow z0 is very small. Can you comment on this in the text? Some find that the difference can be an order of mag- nitude. Were both surfaces at your site particularly smooth? Or could it be something to do with the size of the patch (thinking about the scale/resolution dependency of the microtopographic method – see Fitzpatrick et al. 2019).

**Reply: we have revised this part and give some explanation why ice surface kept at certain domain during melting season, which is related with net shortwave radiation and turbulent heat flux. The former energy item seem increased z0. But turbulent heat flux seems smooth z0.**

Lines 422-425: this paragraph needs rewording so that the first sentence does not seem disconnected from the rest.

**Reply:  we have revised accordingly**

Lines 430-433: this is a significant finding; however, there is something about the word- ing in this sentence that I think should be addressed – as z0 is in this instance (using the bulk method) required to calculate the turbulent fluxes, arguing that the turbulent heat index (calculated with turbulent fluxes) is a determining factor seems circular. I think the statement could be made more clearly, perhaps referring to the association between the two rather than a causal relationship.

**Reply: we have add the profile method and bulk method. Both method shown a similar relationship. A lagged correlation was also applied in the revised manuscript to indicate the**

**relationship between main energy items and z0.**

Line 434: Make sure terminology is clear here – you refer to the August-one ice cap, and then call it a glacier. In my understanding, these are different.

**Reply: we have revised accordingly. "the August-one glacier" changed to "the August-one ice cap" across whole manuscript.**

Line 439: The second sentence can be deleted, it does not add anything to the findings or argument.

**Reply: We have deleted the last sentence. The revised part has changed as" This study found an exponential relationship between $z_0$ and $L_S$. The delicate role of $z_0$ played in the ice surface balance is still not fully known. Further comparative studies are needed to investigate the $z_0$ variation through eddy covariance, profile method and DEM-based $z_0$ estimation."**

Conclusion

I think comparison to other ice masses, and links to other studies/locations should be made in the discussion, with some thought given to whether you might find the same results where ice z0 and snow z0 have greater contrast. And, while it is important to acknowledge the site specificity of a study, further studies are always required and saying so in the conclusions is superfluous. Instead, the main messages from the paper (3 or 4 of them, as far as I can see) should be summarised here.

**Reply: thanks for your suggestions. We have revised accordingly**

**Reply to comments from anonymous referee #2**

The study by J Liu et al. demonstrates the first use of an automated photogrammetric apparatus to monitor surface roughness at a daily timescale for an ice cap in China. The authors supplement these observations from a single site with meteorological records from nearby, as well as manual photogrammetric measurements at a variety of locations across the ice cap during the course of the ablation season. The authors thus investigate spatial and temporal variations of surface roughness during the ablation season, as well as linkages to surface energy balance. From the automated roughness measurements they find that roughness is temporally variable and highly modified by precipitation, with both rain and snow precipitation leading to a reduction in roughness. From the manual measurements, they find that the seasonal firm/ice transition zone corresponds to the maximal surface roughness at any point, while ice or snow surfaces both exhibit lower surface roughness. The authors also suggest a link to the importance of turbulent fluxes in the whole energy balance.

The target of spatially-extensive surface roughness measurements is a novel development, and useful to understand roughness variations. While the general patterns of seasonal and spatial variability are very likely to be accurate and form a nice story, the authors seem to have some fundamental misunderstandings about surface roughness metrics and their meaning. In addition, the methods are not entirely clear, results are given to an unrealistic and misleading precision (also without any uncertainty assessment), and although the written English is generally correct, the writing style is particularly abrupt. Consequently, although the authors have painted a nice picture of the spatiotemporal evolution of surface roughness at August-one ice cap, the manuscript needs substantial revisions before it should be considered for publication in The Cryosphere.

*Major points:*

Fundamental misunderstanding of surface roughness. The authors seems to confuse Z_o and topographic surface roughness, which are not the same: while approaches have linked the two, the aerodynamic roughness length is not simply a topographic parameter, and efforts to assess Z_o based on topographic parameters need to be validated with micrometeorological measurements. Furthermore, the authors' effort to produce a grid-based estimate of surface roughness is only applicable for the case of isotropic roughness, which is not the case for ice surfaces.

**Reply: Thanks for your comments. We have revised the manuscript based on your suggestions and comments. We have revised the topographic surface roughness as aerodynamic surface roughness ($z_0$) accordingly. For spatial and temporal $z_0$ variation, precisely capture wind direction data was not**

**available across the ice cap. In this case, we applied averaged four cardinal direction $z_0$ in the revised manuscript.**

**For the Anisotropy problem in here. We provide DEM based four cardinal direction z0 and Munro (1989) based profile method estimated z0 in Figure 5. Snow and ice surface photos were also provided to shown ice or snow surface features in Figure 5 and Figure 9. Glacier surface did showed some isotropic features. In order to avoid anisotropy in calculate turbulent heat flux, we estimated upwind $z_0$ by consider the prevailing wind direction data at top of ice cap. We have explained in the revised manuscript. The sensible and latent heat was calculated based on 4m half hour meteorological data and daily estimated prevailing wind direction $z_{0\_DEM}$.**

Lack of clarity with regards to several methods. The authors mention two specific efforts to estimate Z_o from topographic profiles: Lettau (1969) and Munro (1989). It is not clear which is actually used in this study, now how it was applied to the gridded height data. In addition, numerous details of the energy balance model used are missing, while the authors may have accidentally disregarded conduction of heat.

**Reply: we have revised and clarify the detrend method and $z_0$ calculation. The Munro method was applied here in the revised manuscript. The subsurface heat flux was also calculated based on 8 level ice temperature observations deep down to 9.25m. The detrended method was presented in Line 566 of the revised manuscript. The upwind prevailing wind direction $z_{0\_DEM}$ was applied to calculate turbulent heat flux in Line 590 to 595. We also provided half-hour meteorological data in Figure 8 instead of daily scale meteorological data.**

Unrealistic precision, no uncertainty of Z_o estimates or energy balance. The accuracy of Z_o is provided relative to control and check points on photogrammetric frames, and is reported to the tenth of a millimetre. However, it is unlikely that the actual measured positions of their control and check points are known to this accuracy. Furthermore, the surface height models produced by the structure-from-motion processing appear to be oversampled by a factor of 10x in each dimension, relative to the reported point densities. Finally, no assessment of uncertainty has been conducted for the Z_o estimates or the energy balance calculations.

**Reply: The accuracy of check points and control points provided here are based on the Agisoft reports which provided precision information for each plot. In the revised manuscript, we have provided precision uncertainty about check points and control points in Figure 4. The oversampled by a factor of 10x in each dimension have revised from 0.1mm to 1mm.**

The uncertainty of Z0 have conducted and provided in the revised manuscript. No evidence given of cryoconite, but of red algae. This may be a misunderstanding of some sort, but the authors refer multiple times

to the development of cryoconite and its effect on surface roughness, a phenomenon that would certainly explain some of the surface roughness dynamics that they observe. However, the first time cryoconite is mentioned is with regards to Figure 2, but Figure 2 does not provide any evidence (to my eye) of cryoconite – rather, red snow algae is clearly evident. This gives some concern of a basic misinterpretation of results.

**Reply: we have provided evidence of photos in Figure 3 and Figure 5 to shown cryoconite. Two photos were also provided here to shown more detailed information about its size and ice surface cryoconite holes over August-one ice cap. Actually in the field work, we sampled surface cryoconite at 20cm *20cm plot, and dried it in the laboratory. Most of the substance was small mineral particles. The cryoconite appears red, it might related with it high concentration of Fe in it (Li et al., 2019).**

**For uncertainty of z0, we provide the uncertainty at Figure 3s, which is the mean of four cardinal direction z0 (Figure 3s a). The mean of Munro profile method calculated z0 was also provided (Figure 3s b). For the uncertainty of prevailing wind direction z0_DEM, we only acquired one data at every data. In this case we do not provide uncertainty in the revised manuscript.**

**Li Y, Kang S, Yang F, Chen J, Wang K, Paudyal R, Liu J, Qin X, Sillanpaa M, Cryoconite on a glacier on the north-eastern Tibetan plateau: light-absorbing impurities, albedo and enhanced melting. Journal of Glaciology, 65(252) 633-644.**

[Figure]

Figure 1 Ice surface cryoconite and cryoconite holes.

Some grammar improvement needed, also some changes to the writing style are needed, as it is not currently suitable for TC.

**Reply: Thanks for your suggestions, we have revised based on the detailed comments.**

*Detailed comments:*

L1-2…during 'the' melting season

**Reply: we agree. Now add 'the' as suggested.**

L18. Zo was calculated from this data – you need to say how. Manual measurements of what type? Micrometeorology? Profiles of elevation difference?

**Reply: We totally agree. This sentence has revised as: '$Z_0$ was estimated based on microtopographic methods from automatic and manual photogrammetric data.'**

L37-63. It is apparent from this section that the authors misunderstand several key concepts relating to Zo and turbulent heat transfer more generally; I suggest a careful read of Smith et al (2016) for a review of the differences. First, $Z_o$ may be commonly called 'surface roughness' but its full title is the 'aerodynamic roughness length' (for momentum transfer/heat transfer). In any case, it emerges in the bulk aerodynamic approach as a constant of integration that results from the interaction of the boundary layer with the surface. It is a meteorological term (not a topographical term) that is influenced by both properties of the boundary layer and the surface.  One can determine an effective surface roughness 'directly' from eddy covariance measurements (and less directly from wind towers), but it is highly variable in time primarily because the boundary layer is often highly variable. The variability of the boundary layer leads to a different fetch over which the layer is interacting with the surface topography. The microtopographic roughness (which you have calculated) is thus a very good indication of $Z_o$, but the relationship is not direct or linear, as the energy balance is controlled not just by surface topography at an individual location, but is variably influenced by its surroundings (e.g. Steiner et al, 2019). Thus, it is difficult to trust the values of $Z_o$ produced by this study, as they are not validated by wind tower or eddy covariance observations (which actually resolve $Z_o$). However, microtopographic roughness metrics are a very strong proxy for $Z_o$ (e.g. Nield et al, 2013), so I have much more confidence in the temporal and spatial variability presented by the authors. However, I think they need to very carefully reframe their introduction to conform with established theory.

**Reply: Thanks for your suggestion and comments. Your comments have greatly help us to revise this manuscript. We have revised 'surface roughness' as 'aerodynamic surface roughness'. The 'direct' or 'indirect measurement' have revised as 'estimated'. Because the location are different between the automatic photogrammetry observation and the wind tower. In this case, we did not shown the calculated z0 based on wind tower data. Actually, we have eddy covariance observations at the ice cap top since 2016. In 2019, we have move the microtopographic observation to the ice cap top in order to carry out comparison with wind tower and eddy covariance.**

L42. Please provide references indicating that microtopographic $Z_o$ is more accurate than wind profile or EC measurements. I don't know how one can claim this, as those methods are the 'ground truth' of $Z_o$ at a site.

**Reply: We made a mistake here. The estimation of z0 based on microtopographic method showed some**

**advantages over EC measurements or profile methods rather than more precise.**

230    **We have revised this part as "However, micro-topographic estimated $z_0$ shows some advantages such as lower scatter than profile measurements over slush and ice (Brock et al., 2006), and easily application at different locations (Smith et al., 2016)"""**

L47. 'Direct measurement' is strange nomenclature; microtopographic approaches, including the Lettau (1969) approach, are anything but direct.

235    **Reply: We fully agree and revised the "direct measurement" as "microtopographic estimated z0". The rest of the paper also changed accordingly.**

L52. Rees and Arnold (2006) is also sensible to mention here.

**Reply: We have add accordingly.**

L55. Other examples of this approach are Rounce et al (2015), Quincey et al (2017), and Miles et al (2017).

240    **Reply:    Thanks for your recommendation, we have added accordingly.** L56. The photogrammetric approaches need validation, as the relationship between topographic roughness and aerodynamic roughness length is also affected by local meteorology (Nield et al, 2013).

**Reply: Thanks for your valuable comments.**

**We have revised this part as 'Such data facilitate the distributed parameterization of aerodynamic**
245    **surface roughness over glacier surfaces (Smith et al., 2016; Miles et al., 2017; Fitzpatrick et al., 2019) Precision of microtopographic estimated z0 also became an major concern, and lots of comparative studies with aerodynamic method (eddy covariance or wind towers measurements) carried out over debris-covered or no-debris covered glaciers. Some of the studies showed the difference was within an order of magnitude (Fitzpatrick et al., 2019) or strongly correlated (Miles**
250    **et al., 2017).'**

L74/Figure1. Both panels need a scale. The political map of China is irrelevant to the current study; of more relevance are dominant weather patterns and elevation, including areas outside China's claimed border. Furthermore there is no need to depict the South China Sea, which results in a very poor use of space. What is the polygon within China? It is not identified in the figure or caption.

255    Please provide information about the image of August-one in panel (b) – date, satellite, etc. Red and green are poor choices of color for icons in panel (b), as many people cannot distinguish between these two colors.
**Reply:We have edited as suggested.**

L79. Please provide sensor specifications and measurement uncertainties for the AWS.

**Reply: Done**

260 **We add Table 1 as :**

**Table 1 Measurement specifications for the AWS located at the top of the glacier (4820 m a.s.l.). The heights indicate the initial sensor distances to the glacier surface; the actual distances derived from the SR50A sensor.**

| Variable | Sensors | Stated accuracy | Initial Height (m) |
|---|---|---|---|
| Air temperature | Vaisala HMP 155A | ± 0.2ºC | 2, 4 |
| Relative humidity | Vaisala HMP 155A | ± 2% | 2, 4 |
| Wind speed | Young 05103 | ± 0.3 m/s | 2, 4 |
| Wind direction | Young 05103 | ± 0.3º | 2, 4 |
| Ice temperature | Apogee SI-11 | ± 0.2ºC | 2 |
| Shortwave radiations | Kipp&Zonen CNR-4 | ± 10% day total | 2 |
| Longwave radiation | Kipp&Zonen CNR-4 | ± 10% day total | 2 |
| Surface elevation changes | Campbell SR50A | ± 0.01 m | 2 |
| Precipitation | OTT Pluvio$^2$ | ± 0.1 mm | 1.7 |

L81. The sensor measures relative surface height; it does not measure mass balance. Also in L104

265 **Reply: We agree. We have revised L81 sentence as' Surface relative height is measured by a Campbell Scientific ultrasonic depth gauge (UDG) close to the AWS'**

**For Line 104. We used a hunting-video camera to take pictures of ice-surface gauge stakes near automatic photogrammetry site. We expect it is mass balance of the site. The Figure 1 shows the hunting camera and stake close to top of the August-one ice cap, and rough surface and stake captured by**
270 **hunting camera at the automatic photogrammetry site on September 9 of 2018. For clarity, we have**

revised the sentence in line 104 as 'Surface elevation changes caused by accumulation and ablation was measured by digital infrared hunting-video camera, which took pictures of ice-surface gauge stakes located near the automatic photogrammetry site.'

[Figure]

 Figure 2 Left side photo shows hunting-camera and mass balance stakes close to top of the August-one ice cap, right side photo showed hunting camera photographed rough ice surface on September 8 of 2018 and ice surface stake gauge at the automatic photogrammetry site.

L83. There was a windbreak fence installed on the glacier?

Reply:    The wind break fence was installed for the OTT Pluvio$^2$ precipitation gauge. For clarity, we
280    have revised as' An all-weather precipitation gauge adjacent to the AWS measures solid and liquid precipitation'.

L94. How were the positions of the control and check points measured? You report accuracies relative to these positions of less than 1 mm, but I am not convinced that you could locate the control point positions to a higher accuracy than this. Also, how was the frame structure anchored?

285    Reply: The report accuracies relative to these positions of less than 1 mm did have some problems. We have revised it and add uncertainty of precision.

For manual photogrammetry, a 1.1×1.1m portable square aluminum frame was applied as control field.

Geo-reference of the point cloud was enabled using control points established by four cross-shaped screws on the four corners of aluminum frame. Four cross-shaped screws on the middle of aluminum rimes used as check points. The location of these screws was measured precisely with millimeter brand tape. The frame structure just put on the ice surface without anchored.

[Figure]

Figure 3 Aluminum frame used as control field for Geo-referece at August-one ice cap. The hummocky is covered by cryoconites (grey part is sun dried cryoconite, brown part is wet cryoconite).

For automatic photogrammetry, a wooden frame, 1.5 m wide, and 2 m long, was put on the ice surface. This frame served as a geo-reference control field (Figure. 2b). The wooden rectangle frame was made by 4 water proofed 3 m rulers. The frame was put on the ice surface and chained together with two aluminum stakes ahead of the automatic photogrammetry camera. The wooden frame stands freely on the glacier and sinks with the melting surface.

All the control points and check point are located at feature points of the wooden frame, and these points also measured very carefully with millimeter brand tape.

L102. Did you choose the daily best-exposed sets of photos manually or automatically? For days with multiple very clear photo sets, was there strong agreement in derived Z_o or a consistent diurnal variation?

Reply: We choose the best-exposed sets of photos manually. Cloudy or frosty weather affected automatic photogrammetry exposures, and heavy snowfalls resulted in a texture-less surface. We choose photos to avoid these bad weathers.

**Detailed analysis of diurnal variation was not carried out yet. Since z0 highly affected by weather conditions, Snowfall, rainfall also affected, and Refreezing at night could also affect the ice surface z0.**

L112. Does the August-one ice cap have an accumulation area?

**Reply: We have observed for the last 5 years. No accumulation area for the ice cap.**

L120. How are the seven pairs of convergent photos arranged? Do you use all 14 photos to produce the DEM and orthoimage? Did you ever carry out the manual photogrammetry at the automatic site?

**Reply: We revised this part as:' Seven to twelve of such photos were taken at each survey site and surrounded the target area from different directions.'**

**We did not carried out manual photogrammetry at the automatic site.**

L124/Figure 4. Panels b and c are switched relative to the text, which led to some confusion about the numbers of check points and control points. I see no evidence of cryoconite in the image, but of red algae which is commonly found on melting snow.

**Reply: We have revised Figure 2.**

**In the revised manuscript, we split Figure 2 into Figure 2 and Figure 3. Figure 2 showed the automatic photogrammetry device. Figure 3 showed the control field and detrend DEM data.**

**In Figure2c, the cryoconite in the image was not clear. We have provide more clear evidence in revised Figure3c. The photo of Figure3c showed cryoconite hummocky, in which top of the mound were dry cryoconites, underneath were wet cryoconites.     The color of cryoconite over August-one ice cap is not red, it is brown color.**

L135. The standard reference for this processing workflow as applied to glaciers is Westoby et al (2012). Also, this approach has already been applied to estimate surface roughness of glacier surfaces: Quincey et al (2017), Miles et al (2017), Steiner et al (2019).

**Reply: Thanks for your suggestions, we have revised and cited these references.**

**We revised as' Structure-from-motion photogrammetry is revolutionizing the collection of detailed topographic data (Westoby et al., 2012; James et al., 2017). High resolution DEMs produced from photographs acquired with consumer cameras needs handled carefully (James and Robson, 2014). In this study, both manual and automatic photographs were imported into a software program, Agisoft Photoscan Professional 1.4.0. This software allowed us to estimate camera intrinsic parameters, camera positions, and scene geometry. Agisoft Photoscan Professional is a commercial package which implements all stages of photogrammetric processing (James et al., 2017). It has previously been used**

**to generate three-dimensional point clouds and digital elevation models of debris-covered glaciers (Miles et al., 2017; Quincey et al., 2017), ice surfaces and braided meltwater rivers (Javernick et al., 2014; Smith et al., 2016). After new snowfall, it was difficult to match feature points in the photo sets. Three days of automatic data could not be processed. We estimated z0 data for the missing days based on data from snowfall days at the automatic site.'**

L141-146. This content belongs in the background. Note that Lettau (1969) was the first such effort (of which I am aware). It is also worth noting the extensive review of microtopographic metrics by Nield et al (2013).

**Reply: We have revised this part. The title of '2.5 Roughness calculation' has revised as '2.5 Aerodynamic roughness estimation'**
**The content of this paragraph has revised and referenced Lettau (1969) and Nield et al. (2013).**

L145. Munro (1989) is probably the appropriate first reference here, as is Brock et al (2006).

**Reply: We revised and add these two references accordingly.**

L161. The method described (based on the standard deviation of detrended elevation) is precisely the Munro (1989) method.

**Reply:   In the revised manuscript, we referenced the Munro (1989) method.**

**We have revised as' Based on the work of Lettau (1969), Munro (1989) simplified the equation (1) by assuming that h\* can equal twice the standard deviation of elevations in the de-trended profile, with the profile's mean elevation set to 0 meter. The aerodynamic roughness length for a given profile then becomes'**

L172-3. Averaging over cardinal directions is only meaningful for surfaces that are isotropic. However, the literature has repeatedly shown that melting ice is strongly anisotropic, as the direction of wind strongly dictates the pattern of melt, and feeds back via roughness. So this 'averaging all cardinal profiles' is entirely unsuited to your study site, unless you can demonstrate that the ice surface is indeed isotropic in terms of roughness, which would be highly surprising.

**Reply: We agree.**

**We have revised it as ' For manual photogrammetry, we put the aluminum frame horizontally over the ice surface, the plot is detrended by setting the control points at z axis of the same values. For automatic photogrammetry, the control field of wooden frame was also laid horizontally over the ice surface that lowered as the ice melted and maintained a horizontal position between the control field and ice surface. A DEM based approach enables the roughness frontal area s to be calculated directly for each cardinal wind direction (Smith et al., 2016). The combined roughness frontal area was calculated across the plot,**

**the ground area occupied by micro-topographic obstacles is 1m2. We used a DEM-based average**
**$(\bar{z}_{0\_DEM})$ of four cardinal wind directions to represent overall aerodynamic surface roughness. Based**
**on the half-hour wind direction data at the August-one ice cap, the daily upward wind direction DEM-**
**based z0_DEM was also estimated at the automatic photogrammetry site. Considering that wind**
**direction changed during the day, in this case we selected the prevailing wind direction to calculate**
**frontal area s. The prevailing upwind direction DEM-based z0_DEM was applied to calculate turbulent**
**heat flux. Using the Munro (1989) method, z0_Profile was calculated for every profile (n=1000) in both**
**orthogonal directions for each plot at the automatic photogrammetry site.'**

L174. Some things are not entirely clear to me about your method. First, do you use all profiles in each cardinal direction? Second, it is not clear if you have implemented the exact Lettau approach or the Munro approximation in your 'all profiles' approach. Third, such an implementation (all profiles averaged, for either Lettau or Munro) has already been implemented and tested for a glacier surface. Please see Miles et al, (2017).

**Reply: We have used Lettau method for DEM based method, Munro method for profile method. The**
**results was presented in Figure 5a. Average of four cardinal direction ($\bar{z}_{0\_DEM}$) and average of profile**
**method (z0_Profile), the prevailing wind direction z0_DEM    was all presented in Figure 5. z0_DEM**
**was applied to calculate turbulent heat flux.**

**In the revised manuscript, we give detailed description in line 185 to 195, and line 210-215.**

L179-181. The surface energy balance presented is not quite accurate for a 'melting' glacier, but for a 'temperate' glacier. Do you have any evidence that the August-one ice cap is temperate? If not, there also needs to be a term for heat conduction.

**Reply: we have revised and add subsurface heat flux (QG) based on the observations at the ice cap. We**
**have a subsurface temperature observation at five different depth. The maximum depth was 9.25m**
**(beginning in 2015).**

**We have revised as 'The subsurface heat flux QG is estimated from the from the temperature-depth**
**profile and is given by $Q_G = -k_T \frac{\partial t'}{\partial z'}$ where kT is the thermal conductivity, 0.4Wm-1K-1 for old snow**
**and 2.2W m-1K-1 for pure ice (Oke, 1987).'**

**The result of $Q_G$ was presented after rainfall energy in the revised manuscript as' Compared to**
**other energy components, QG was very small, with a daily mean of -0.65 W m-2 and a maximum and**
**minimum of -0.4 and -2.1 W m-2, respectively.'**

L191. My impression is that you use your calculated Z_o value for the bulk aerodynamic approach. How do

400  you integrate your 3-hourly (half-day) Z_o values with your model? At what timescale is the model run? What uncertainty does the input meteorology have, and what uncertainty does this produce for your results?

**Reply: The turbulent heat flux was calculated based on half-hour meteorological data at 4m level and daily scale $z_{0\_DEM}$. We assumed the z0 was same in each day. In the revised manuscript, we have revised it**

405  **We revised as 'In a horizontally homogeneous and steady surface state, the surface heat fluxes $Q_E$ and $Q_H$ can be calculated using either the bulk aerodynamic approach or profile method, based on the Monin-Obukhov similarity theory (e.g., ; Arck and Scherer, 2002; Garratt, 1992; Oke, 1987). In this study, half-hour observations at 4 m level and daily upward wind direction DEM-based $z_0$ were used to calculate $Q_E$ and $Q_H$ based on the bulk method.'**

410  L204. Is an environmental lapse rate entirely appropriate for this site? Do you have lapse rate measurements?

**Reply: We do not have lapse rate measurement here at the ice cap. We have applied temperature lapse rate of 5.6 °C $Km^{-1}$ observation results not far from here by Chen et al. (2014).**

**We have revised as 'In order to calculate $P_r$, we used the air temperatures recorded at the AWS. There is an elevation difference between the study site (4700 m) and the AWS (4790m); recorded air**
415  **temperatures were corrected to account for the elevation difference, a lapse rate of -5.6 °C $Km^{-1}$ was applied based on observation nearby (Chen et al., 2014)'**

L205. How confident are you that the AWS measurements are broadly representative of the entire ice cap? Do you have evidence to back up this claim?

**Reply: the ice cap is flat and open terrain as shown in Figure1. The AWS is only 1500m away from the**
420  **automatic photogrammetry site and 90m difference in altitude. This topographic feature favors the representative of the AWS over the ice cap.**

**We have revised this as' The ice cap is flat and open terrain so in this case wind speed and relative humidity at the study sites were assumed to be close to those observed at the AWS.'**

L210-211. I am not sure how you get seventeen (17), as you have 4 control points and 3 check points.
425  Similarly, I do not understand what the 31 manual photography pairs are – please explain.

**Reply: we have revised as' We used seventeen plots to analyze the horizontal and vertical accuracy of our automatic photogrammetry, and thirty-one plots for our manual photogrammetry'**

L210-219. This entire section is an amalgamation of bullet points; please rewrite to conform to style for The Cryosphere.

**Reply: We have revised it.**

L212 and L216. The reported point densies do not justify a resolution of 0.1mm, but of 1mm. These DEMs are 100x oversampled.

**Reply: Agree, We have revised it accordingly.**

L213. The average georeferenced error is greater than 1mm for half of the control points, and nearly all check points. However, I am also not certain how precisely you could have measured the location of the control and check points. Please provide details and uncertainty.

**Reply: We have provide details at L94 for measurement of the control and check points. We aslo provide uncertainty for check points and control points in Figure 4.**

L225. Yes, but part of this is also the difference of your survey design. For the automatic measurements, the camera is moving linearly, and the density of tie-points is much higher in the foreground compared to the background. For the manual method, although the survey design is not clear, more photos were taken and I presume that they surrounded the target area. This type of survey would be expected to provide a much more robust elevation model.

**Reply: We agree. We have revised the manual survey was different from automatic photogrammetry. The manual survey surrounding the target, and automatic measurements moving linearly.**

**We have add this difference here and revised as' Note that the control and check point errors were larger for the automatic measurements than for the manual ones (See Figures 4). We believe that this is the case because, rather than using static f-stop and exposure times (as in automatic photogrammetry) researchers engaged in manual photogrammetry could adjust exposure time based on ice surface conditions. This allowed production of better quality photos even on cloudy or foggy days. The difference of survey design also caused more precise results for manual than automatic photogrammetry. For the automatic measurements, the camera was moving linearly, and the density of tie-points was much higher in the foreground compared to the background. For the manual method, photos were taken by surrounding the target area. This type of surface provided a much more robust elevation model and points density.'**

L228. Rees and Arnold (2006) did indeed suggest millimetre vertical accuracy. The also suggest a fetch length of 3-6 meters as relevant for the majority of energy balance situations, which is considerably larger than your domain.

**Reply: we have revised it. Rees and Arnold (2006) suggested millimeter vertical accuracy only for 1D**

profile, not for 2D DEM data. The suggest a fetch length depend on the topography. In this study, a 1m square are more portable for manual photogrammetry. A lager plot scale need put camera 5-to 10 m or much higher locations to catch larger scale ice surface $z_0$. In this study we do not include larger plot scale comparative studies.

L230/Figure3. It is not clear what this chart shows – the y axis is labelled 'Differences', but is this RMSE, MAD, or…? Please clarify.

**Reply: We have revised it as 'Standard derivation'.**

L231-4/Figure 4. Same problem and Figure 3. Should be merged with Figure 3 as a second panel.

**Reply: we have merged with Figure3, and revised as 'standard deviation'.**

L238. No description of profile analysis is included in the manuscript, only of a DEM analysis. Please provide more detail.

**Reply: We have provided more detail about profile results.**

L239. Do you have an estimate of the uncertainty of these Z_o values?

**Reply: For the average of four cardinal direction $\bar{z}_{0\_DEM}$ and Munro profile method calculated average of z0_Profile, We provide uncertainty in Figure 3s. For prevailing wind direction Z0_DEM. We do not have uncertainty because we have one data every day.**

L242-254. Listing a narrative as bullet points in the results is not particularly aesthetic, and this section should be rewritten as a paragraph. More importantly, this section mixes results and interpretations. Please present the observations, then interpret them.

**Reply: We have revised it as: 'At the start of the observation period of July 12, snow covered the study site. As the snow melted, the ice cap surface z0 increased. During this periods, z0 dropped to around 0.1mm due to intermittent snowfall. On July 21, cryoconite appeared on patches of snow-crust, which led to patchy melt. From July 21 to 24, overall z0 increased from 0.1mm to 1.6mm. By July 29, snow had disappeared from the study site, and z0 fluctuated but trended lower. From July 29 to August 5 bare ice covered whole field of view, and ice surface z0 ranged from 0.18 to 0.56mm. From August 6 to September 3 there was intermittent snowfall followed by melting, z0 ranged from 0.1 to 1.0mm. From September 4 to September 14 z0 showed an overall increase, reaching a maximum of 2.5 mm on September 8. There was intermittent snowfall during this period, which temporarily reduced z0. Z0 then increased due to patchy micro-scale melting. After September 14, snow covered the whole surface**

490 **of the ice cap. There was no melting and little fluctuation in z0.'**

L254/Figure 5. Z_o values are more commonly presented on a logarithmic scale, as even a factor of 2 makes little difference in the turbulent fluxes, whereas a factor of 10 can be a considerable difference regardless of value. This is, in part, due to the bulk aerodynamic approach. Also, it would be very nice to include a set of panels depicting the surface at different parts of this record (high and low values, for example).

495 **Reply: Thanks for your suggestions, we have revised accordingly.**

L258. One order of magnitude is not a particularly large variation of Z_o.

**Reply: We have revised as' It should be clear that $z_0$ varied from 0.05 to 2.74 during melting season'**

L260. I have not yet seen evidence of cryoconite holes; the image in Figure 2 is unconvincing. Also applies to L280

500 **Reply: We have add surface photos from July 12 to September 13 to show ice surface features at different periods in revised manuscript of Figure 5.**

L263,276,277. I see no need to include p-values here.

**Reply: we have revised it.**

L274. Was there no accumulation in this year?

505 **Reply: In August, at top of the ice cap, the mass balance is**

**already negative.**

L283/Figure 6. Is there a reason that the lines are shown with different styles? For comparison, it would be good for all 4 panels to have the same y-axis limits.

**Reply: we have revised it.**

510 L288-302. Somewhere in this section there should be a reference to Figure 7.

**Reply:   we have revised and referenced Figure7**

L310,324,353,339,345. The use of sub-headings here just breaks up the text.

**Reply: We have deleted these sub-headings.**

L320/Figure 7. In panel (a), please use a logarithmic scale for Z_o. Is panel (c) showing net solar radiation, or downwelling – not specified. The y-axis upper limit in panel (d) should be 100%. In general, all time-series look smoothed. Please provide details of exactly what is shown. In the caption, please be sure to provide the year.

**Reply: we revised it accordingly.**

L330/Figure 8. What is the uncertainty of each of these values quantities?

**Reply: The Figure 8 displayed daily main energy items in which net radiation is calculated based on half hour observation of net shortwave radiation and net shortwave radiation. Latent and sensible heat is calculated based on half hour meteorological and daily windward direction DEM- estimated $z_{0\_DEM}$ (we assumed z0 is not changed during the day). The uncertainty is not included for simplicity. We have provide half-hour scale latent heat documents.**

L335. If latent heat and sensible heat account for so little of the energy balance, how much impact does a variation of Z_o from 0.25 mm to 2.5 mm have on the total energy balance?

**Reply: In this study, we calculated latent heat and sensible heat based on the bulk method. We do not include sensitive test of z0 variation on total energy balance. Actually we applied $z_{0\_Prifile}$, $z_{0\_DEM}$ and $\bar{z}_{0\_DEM}$ to the bulk method. Highly constant results was acquired between three different z0 (Figure 4s).**

L343. The 'visible smoothing' is not clear to me from Figure 7. Please explain where you see this.

**Reply: we have revised it and added two ice surface photos before rainfall and after rainfall event in Figure 9 to indicate the smoothing process.**

L349-350. As turbulent fluxes matter very little for your energy balance, the match is not due to the calculated Z_o.

**Reply: we agree. We have revised it accordingly.**

L358-380/Fig10 and 11. I do not think this analysis is very well grounded in theory. First of all, as the turbulent fluxes depend on Z_o, you are comparing a quantity to a modified version of itself in Figure 10d and Figure 11. In fact, this exactly corresponds to the shape of the fit in bulk aerodynamic theory (which you have used to relate Z_o to the turbulent fluxes). So on one hand, none of this section is unexpected, but nor does it provide any novel insight. On the other hand, if you intend to examine the potential feedbacks between energy balance and surface roughness, that would be very interesting, but would require the use of a lagged correlation (in which case your variables would be independent).

**Reply: Thanks for your excellent question and suggestions. We totally agree with you since Figure 11 and 12(revised manuscript) comparing DEM_ based $z_0$ with turbulent fluxes which were calculated based on it. In this case we have compared DEM-based $z_{0\_DEM}$, $z_{0\_Profile}$, and $\bar{z}_{0\_DEM}$ with bulk method calculated turbulent fluxes to avoid the problem you mentioned. We also compared $z_{0\_Profile}$, and $\bar{z}_{0\_DEM}$ and main energy items, which all have similar results (Figure 1s and Figure 2s).**

**We have revised in method part as' Figure 11 shows the relationship between daily upward wind direction DEM-based z0_DEM and the main energy flows. Scatter diagrams showed a positive relationship between z0 and net shortwave radiation (Figure 11a, r=0.1) and a significant negative relationship between z0 and net longwave radiation (Figure 11b, r=-0.35), Graphing z0 vs. bulk method estimated latent heat showed a significant negative exponential relationship (Figure 11d, r= -0.35). The scatter diagram showed no significant relationship between z0_DEM and the bulk method estimated sensible heat (Figure 11c). The average of the Munro profile based z0_profile and DEM based $\bar{z}_{0\_DEM}$ and the main energy items are also analyzed respectively. Scatter diagrams showed significant negative relationship between z0_profile and net longwave radiation (Figure1s b, r=-0.5). Graphing z0_profile vs. the bulk method estimated sensible heat showed a significant negative exponential relationship (Figure 1s d, r=-0.69). These scatter diagrams showed no significant relationship between z0_Profile and the bulk method estimated sensible heat (Figure 11c, 11e). $\bar{z}_{0\_DEM}$ vs. the bulk method estimated latent heat showed a significant negative exponential relationship (Figure 2s d, r= -0.44). The scatter diagrams between $\bar{z}_{0\_DEM}$ and net shortwave radiation, the bulk method estimated sensible heat showed no significant relationship. '**

L389. Again, Westoby et al (2012) is probably an even more appropriate reference here.

**Reply: Thanks for your suggestion, we have revised and referenced Westoby et al (2012)**

L391. I disagree with this because your survey setup is entirely different for the manual and automatic methods. See my comment with regards to L225.

**Reply: We agree**

**We have revised it as 'We used both automatic and manual photogrammetric methods to sample spatial and temporal $z_0$ variation at the August-one ice cap. Adjust exposure time based on ice surface conditions and survey design of surrounding the target made the manual photogrammetry more precise than automatic photogrammetry (Tables 1 and 2). However, precision is not always the major concern. The glacier surface was a harsh, even punishing environment for the researchers doing manual photogrammetry. In addition, manual photogrammetry took much longer. Automatic methods reduced**

**hours of field work, spared researchers, and produced nearly continuous data. Cloudy or frosty weather affected automatic photogrammetry exposures, and heavy snowfalls resulted in a texture-less surface. Nevertheless, it is likely that photogrammetry techniques will continue to improve and that these drawbacks may be mitigated.'**

L400. I believe you are referring to the glacier terminus. Please replace 'terminal' with 'terminus' throughout the manuscript.

**Reply: We have revised it.**

L403. This is the very interesting result of your study: following the zone of maximum roughness as it migrates upglacier. But a key question is how important are turbulent fluxes in this zone? Perhaps they are relatively unimportant everywhere else, but in this transition zone you have maximum $Z\_o$ and the zone also migrates across much of the glacier, highlighting the importance of transient surface characteristics.

**Reply: Thanks for your comments. We have revised it based on your suggestions.**

L429. Please be careful and consistent with the terminology that you use. In this study you have examined topographic roughness and the aerodynamic roughness length (which are not quite the same thing, see Smith et al, 2016).

**Reply: we have revised topographic roughness as aerodynamic surface roughness.**

L431. I do not think this is a meaningful result, see my comment on L358-380. This also applies to L439.

**Reply: We have revised accordingly**

L434. What do you mean by 'heavy-loading glacier'? I have not heard the term before.

**Reply: we have revised it as' The August-one ice cap dust concentrations are high in melting season.'**

L437. The link between cryoconite holes and surface roughness is indeed important, and you should make this link explicit earlier. However, your manuscript has not presented any clear evidence of the cryoconite development process occurring at your site.

**Reply: We have provided ice surface photos to indicate this processes in Figure5.**

L440. I do not understand what you are referring to here, with regards to quantitative vs qualitative research. Please explain more clearly what you are implying.

**Reply: We have revised it**

L456. What type of studies? Please make some concrete suggestions; at present this discussion and conclusion makes very little contribution to the field.

**Reply: We have revised it accordingly.**

L470 and L472. Duplicate reference.

**Reply: We have revised it accordingly.**

[revised manuscript text omitted]

**Table 2 3 Check point RMSE for manual and automatic photogrammetry**

[revised manuscript text omitted]